# Bayesian extreme value analysis of extreme sea levels along the German Baltic coast using historical information

Leigh R. MacPherson[1], Arne Arns[1], Svenja Fischer[2], Fernando J. Méndez[3], and Jürgen Jensen[4]

[1]Faculty of Agricultural and Environmental Sciences, University of Rostock, Rostock, 18059, Germany
[2]Institute of Hydrological Engineering and Water Resources Management, Ruhr-University Bochum, Bochum, 44801, Germany
[3]Departamento de Ciencias y Técnicas del Agua y del Medio Ambiente, E.T.S.I de Caminos, Canales y Puertos, Universidad de Cantábria, Santander, 39005, Spain
[4]Research Institute for Water and Environment, University of Siegen, Siegen, 57076, Germany

**Correspondence:** Leigh R. MacPherson (leigh.macpherson@uni-rostock.de)

**Abstract.** Developed coastlines require considerable investments into coastal protection measures to mitigate the effects of flooding caused by extreme sea levels (ESLs). To maximise the effectiveness of these measures, accurate estimates of the underlying hazard are needed. These estimates are typically determined by performing extreme value analysis on a sample of events taken from tide-gauge observations. However, such records are often limited in duration and the resulting estimates may be highly uncertain. Furthermore, short records make it difficult to assess whether exceptionally large events within the record are appropriate for analysis or should be disregarded as outliers. In this study, we explore how historical information can be used to address both of these issues for the case of the German Baltic coast. We apply a Bayesian Markov-chain Monte-Carlo approach to assess ESLs using both systematic tide-gauge observations and historical information at seven locations. Apart from the benefits provided by incorporating historical information in extreme value analysis, which include reduced estimate uncertainties and the reclassification of outliers into useful samples, we find that the current tide-gauge records in the region alone are insufficient for providing accurate estimates of ESLs for the planning of coastal protection. We find long-range dependence in the series of ESLs at the site of Travemünde, which suggests the presence of some long-term variability affecting events in the region. We show that ESL activity over the full period of systematic observation has been relatively low. Consequently, analyses which consider only this data are prone to underestimations.

## 1 Introduction

Extreme sea levels (ESLs) and their associated probabilities of exceedance have long been studied due to their role in driving coastal flooding. The application of extreme value analysis (EVA) in this field is thus a well-developed science (Coles, 2001) and best practices based on observations from tide-gauge data have been suggested (Arns et al., 2013; Haigh et al., 2010). However, direct approaches to EVA require sufficiently long records of systematic data to maintain manageable uncertainties at high return periods (Pugh, 2004). In fact, uncertainties in the estimates of ESLs are a major source of uncertainty in expected flood damages in the short term (before 2040; Rohmer et al., 2021), leading to inefficient coastal adaptation. Furthermore, concerns

regarding the sensitivity of estimates to extraordinarily large events have been raised (Dangendorf et al., 2016; MacPherson et al., 2019). Due to the difficulty of including such events in EVA using direct approaches, they are often treated as outliers and disregarded (Hofstede and Hamann, 2022; Jensen et al., 2022). However, excluding errors in reporting or measurement, or realisations of different random processes, these events offer important information on the underlying distribution, especially at the tails, and should thus be kept (Mazas and Hamm, 2011).

Alternatives to direct approaches include the joint probability method (JPM) and regional frequency analysis (RFA). The former was introduced to address the main limitations of the direct methods (Pugh and Vassie, 1980; Tawn et al., 1989; Tawn, 1992) and involves analyses of the astronomical tide and non-tidal residual water level separately, whereby the final probability distribution is obtained based on the joint probabilities of the two components (Haigh et al., 2010). The latter increases the available sample of extremes by combining records within homogenous regions (Weiss et al., 2014; Arns et al., 2015; Bardet et al., 2011) and dealing with local characteristics using a scaling factor. Although both methods address the limitations of direct EVA approaches, concerns regarding their use remain. Principally, they are still constrained by the observation period of the tide-gauges used.

Tide-gauge data offers the most valuable information on ESLs, owing to their widespread implementation, providing systematic measurements often over many decades. In addition to this data, information on ESLs that occurred prior to the introduction of tide-gauges is available at many locations. Despite this, historical events are rarely considered in EVA due to difficulties in reconciling the historical information with systematic data. The main cause of these difficulties is that historical information does not have a well-defined period of observation (Prosdocimi, 2018). That is, traditional EVA depends on a known period in which all sampled extremes have occurred, and as historical measurements are isolated data points, a duration of observation is not defined (Frau et al., 2018). Despite this, several statistical methods exist to combine historical information and systematic data in the field of hydrology (Benito et al., 2004). Of these methods, Bayesian techniques offer a natural framework for handling uncertainties in an extreme value setting (Coles et al., 2003).

In response to gross underestimations of predicted rainfall events in comparison to historical measurements, Coles et al. (2003) introduced a Bayesian approach as an alternative to standard statistical tools for the prediction of extreme events. This was later adapted for flood frequency analysis (Reis and Stedinger, 2005) and the modelling of ESLs (Coles and Tawn, 2005). Here, historical information is treated as censored observations and incorporated into the model likelihood. Standard techniques to maximise the likelihood of such models are intractable, therefore stochastic algorithms such as Markov chain Monte Carlo (MCMC) are employed. This method of modelling extremes, particularly for the incorporation of historical information, has been applied regularly in the field of hydrology (Bulteau et al., 2015; Isikwue et al., 2015; Payrastre et al., 2011; Gaume, 2018; Gaál et al., 2010).

In this study, we apply a Bayesian MCMC algorithm to incorporate historical information in the analysis of ESLs along the German Baltic coast. The region has a long history of ESLs, which includes an extraordinary event occurring in November of 1872 (Jensen et al., 2022). Due to the exceptional magnitude of the event, which lead to widespread flooding, it is often disregarded as a statistical outlier (Hofstede and Hamann, 2022). With the recent devastating floods in western Germany of July 2021 (Mohr et al., 2023; Ludwig et al., 2023), consideration of historical information has grown in relevance (Jensen

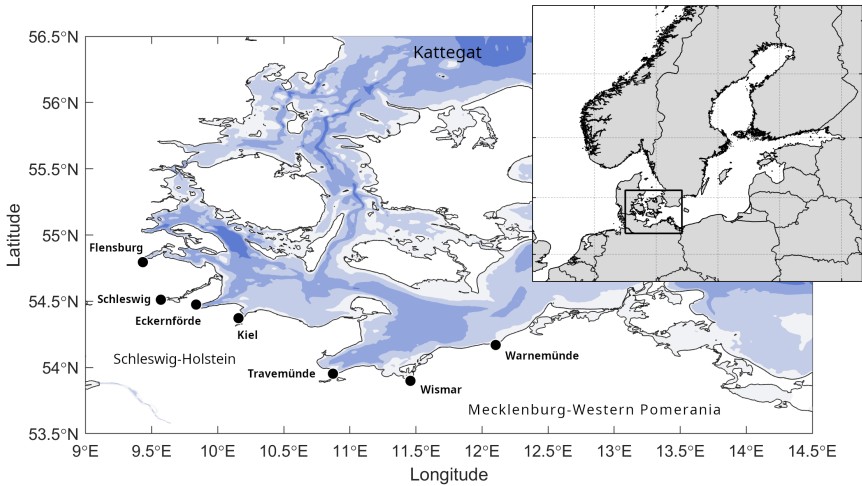

**Figure 1.** Sites considered in this study along the German Baltic coast. Black dots mark the locations of tide-gauges from which systematic records have been used. The two federal states that share the German Baltic Sea coast, Schleswig-Holstein and Mecklenburg-Western Pomerania, have a border at Travemünde.

et al., 2022). We assess ESL quantiles at seven sites along the German Baltic coast where historical information in combination with systematic water level records are available. In particular, we compare estimates of design water levels where historical information is both included and omitted. Lastly, we study recent trends in ESL activity in the region and examine how 60 historical information can provide stable ESL estimates despite short systematic records.

## 2 Background

### 2.1 Study site and data

The German Baltic coast located in the southwest Baltic Sea (Figure 1) has a long history of ESLs. Due to its location, ESLs in the region occur mainly during periods of strong northeasterly winds. However, as the Baltic Sea is a semi-enclosed basin, 65 water levels at the German coast are affected by seiches acting over the entire sea. These seiches are influenced not only by large-scale atmospheric winds and pressure, but by specific sequences of regional wind patterns (Jensen and Müller-Navarra, 2008). Rosenhagen and Bork (2009) and Bork et al. (2022) describe one such sequence in November of 1872, when strong southwesterly winds drove large volumes of water into the Baltic Sea via the Kattegat and caused high water levels in the eastern Baltic Sea. The storm than reversed direction and intensified, causing high water levels and widespread flooding along 70 much of the western Baltic Sea coast. The resulting ESLs along the German coast remain the highest on record, registering approximately 3.4 m above mean sea level (MSL) at Travemünde.

**Table 1.** List of sea level data used in this study, including information on data type, sampling rate and source.

| Data | Type | Sample Rate | Source |
|---|---|---|---|
| Tide-gauge records | Systematic | hourly | GESLA 3 (Haigh et al., 2022), Kelln et al. (2017) |
| AMAX water levels | Systematic/Historical | annual | MLUV (2009) |
| Historical information | Historical | - | Jensen and Töppe (1990), Jensen et al. (2022) |

Coastal defence heights along the German Baltic coast are defined by the two federal states which share the coastline, Schleswig-Holstein (MELUND 2022) and Mecklenburg-Western Pomerania (MLUV 2012). Current design heights are based on an ESL with a return period of 200 years (hereafter referred to as HW200) and include an additional 50 cm to account for future climate induced changes. Values of HW200 are determined based on the statistical analysis of past observations, whose accuracy are thus dependent on the length and quality of the sea level records used. Unfortunately, the exact data and methods used to derive the official return water levels are not published. MLUV (2012) have stated that the Gumbel distribution generally results in the best fit which suggests the use of AMAX data as input.

At present, more than 45 high-resolution (at least hourly samples) tide-gauge records cover the approximately 2,110 km length of the German Baltic coastline (MacPherson et al., 2019). The longest high-resolution (hourly sampling) systematic record was installed at Travemünde at the end of 1949. In addition to these, records of annual maxima (AMAX) water levels are available at 14 sites, which were compiled by the ministry of Agriculture and Environment in Mecklenburg-Western Pomerania (MLUV 2012) from both systematic tide-gauge data and historical information (see Figure 3). These data cover only the coastline of Mecklenburg-Western Pomerania but also at Travemünde, which is a special case due to its location at the border of both states. Here, values for AMAX water levels are available for 184 years between 1826 and 2009.

Lastly, historical measurements can be found for a number of past ESLs at sites located along the Schleswig-Holstein coast. These measurements have been compiled and summarised by Jensen and Töppe (1990). Additionally, Jensen et al. (2022) conducted and in-depth review of historical water levels at Travemünde, providing best estimates of historical ESLs including measurement uncertainties. Where available, we employ all data mentioned above at the sites of Flensburg, Schleswig, Eckernförde, Kiel, Travemünde, Wismar and Warnemünde (see Figure 1 for locations). Sources for the data used in this study are summarised in Table 1, and the extent of the data at each site is illustrated in Figure 2.

## 2.2 Extreme value models

Observed ESLs exhibit an asymptotic behaviour that can be modelled using EVA (Coles, 2001). The choice of extreme value model is dependent on the behaviour of the distribution, which is influenced by how the extreme events were sampled. The two most common extreme value models used in hydrology are the generalized extreme value (GEV) and generalized Pareto (GP) distributions. The former is suited to modelling extremes sampled using the block maxima method, where maxima are taken from individual blocks of data equal in length. For ESLs, which are influenced by seasonal trends, a block length of 1

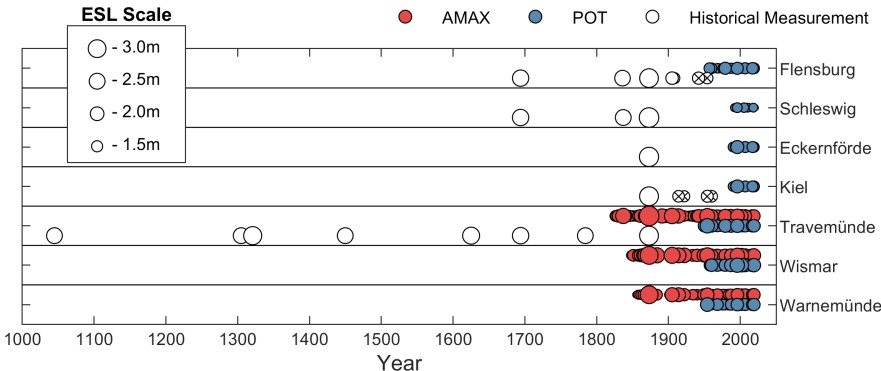

**Figure 2.** The extent of data available at all locations. Each circle denotes a sampled ESL with its size proportional to the event's magnitude (height above NHN). All data has been detrended using MSL. Historical events which lie below the perception threshold and are thus disregarded in the final analysis are shown with a black cross.

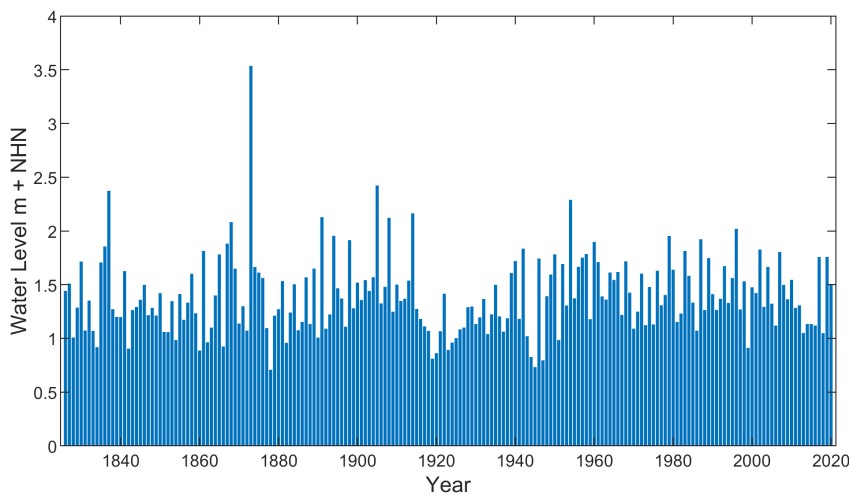

**Figure 3.** Annual Maxima (AMAX) sea levels recorded at Travemünde detrended using MSL.

year is most appropriate, and hence annual maxima (AMAX) samples are preferred. A special case of the GEV distribution occurs when the shape parameter ($\xi$) is equal to 0. Here, the distribution becomes a Gumbel distribution, which is mentioned by
100 MLUV (2012) as the best fit for ESLs along the Mecklenburg-Western Pomeranian coast. An alternative approach is to sample ESLs using the peaks-over-threshold (POT) method, where all events above a certain threshold are selected and modelled using the GP distribution. This approach is generally preferred over the simpler AMAX method as it addresses two main limitations. First, the AMAX method can be wasteful, discounting extremes if multiple large events lie within any one block. Second, it is possible that the analysis becomes biased by the inclusion of moderate values if the data contains long periods of non-extremes.

In this study, we make use of both GEV and GP distributions to model the occurrence of ESLs. Both distributions employ a shape ($\xi$), scale ($\sigma$) and location ($\mu$) parameter. To simplify the probability density function (PDF) and cumulative distribution function (CDF) of both models, we consider the standardized variable $z = (x-\mu)/\sigma$, where $x$ is the independent and identically distributed (iid) random variable to be modelled. The PDF of the GEV distribution is:

$$f(z;\xi) = \begin{cases} \exp(-z)\exp(-\exp(-z)) & \text{for } \xi = 0 \\ (1+\xi z)^{-(1+1/\xi)}\exp\left(-(1+\xi z)^{-1/\xi}\right) & \text{for } \xi \neq 0 \text{ and } \xi z > -1 \\ 0 & \text{otherwise,} \end{cases} \tag{1}$$

and the CDF is thus:

$$F(z;\xi) = \begin{cases} \exp(-\exp(-z)) & \text{for } \xi = 0 \\ \exp\left(-(1+\xi z)^{-1/\xi}\right) & \text{for } \xi \neq 0 \text{ and } \xi z > -1 \\ 0 & \text{for } \xi > 0 \text{ and } \xi z \leq -1 \\ 1 & \text{for } \xi < 0 \text{ and } \xi z \leq -1. \end{cases} \tag{2}$$

The GEV distribution is valid for $x > \mu - 1/\xi$ where $\xi > 0$, and $x < \mu - 1/\xi$ where $\xi < 0$. Where $\xi = 0$, support for the distribution is found for all real values of x. The GP distribution is valid for all $x > \mu$, unless $\xi < 0$, in which case $\mu \leq x \leq \mu - \sigma/\xi$. The PDF of the GP distribution is:

$$f(z;\xi) = \begin{cases} (1+\xi z)^{-\frac{\xi+1}{\xi}} & \text{for } \xi \neq 0 \\ \exp(-z) & \text{for } \xi = 0, \end{cases} \tag{3}$$

and the CDF of the GP distribution is:

$$F(z;\xi) = \begin{cases} 1-(1+\xi z)^{-\frac{1}{\xi}} & \text{for } \xi \neq 0 \\ 1-\exp(-z) & \text{for } \xi = 0. \end{cases} \tag{4}$$

## 3  Methods

### 3.1  Bayesian framework

If we consider a sample of ESL observations $x$, the distribution of events within this sample can be represented using an extreme value model with parameter vector $\theta$. Due to the finite sample of extremes, the exact parameters cannot be known for certain. However, Bayes' theorem relates the posterior distribution of the parameters $\theta$ given the sample of events x, to the likelihood function $\mathcal{L}(x \mid \theta)$:

$$f(\theta \mid x) = \frac{\mathcal{L}(x \mid \theta)f(\theta)}{f(x)}, \tag{5}$$

where $f(\theta)$ is the prior distribution and $f(x)$ is a normalising constant dependent on the sampled extremes only. While including information in the prior distribution can improve inference results and reduce uncertainties (Reis and Stedinger, 2005), El Adlouni and Ouarda (2009) cautions that the choice of prior distribution can introduce bias and should be made with care. Where little to no information on the prior distribution can be found, a non informative prior should be used (Payrastre et al., 2011). In this case, $f(\theta) \propto 1$ and the posterior distribution becomes proportional to the likelihood. By sampling the posterior distribution using a stochastic MCMC algorithm (described in Section 3.2), calculation of the normalising constant is not required.

Formulation of the likelihood function is dependent on the characteristics of the observations, and can be split into two key parts, separating the periods of systematic and historical observations. For the set of $s$ systematic observations $X = \{x_1, x_2, ..., x_s\}$ and $h$ historical observations $Y = \{y_1, y_2, ..., y_h\}$, the likelihood function is:

$$\mathcal{L}(X, Y \mid \theta) = \underbrace{\mathcal{L}(X \mid \theta)}_{\text{syst. likelihood}} \cdot \underbrace{\mathcal{L}(Y \mid \theta)}_{\text{hist. likelihood}} . \tag{6}$$

The likelihood function for the systematic data is simply equal to the product of the probability density ($f_\theta$) of each observation:

$$\mathcal{L}(X \mid \theta) = \prod_{i=1}^{s} f_\theta(x_i). \tag{7}$$

Formulation of the historical likelihood function is more complex and is taken from Payrastre et al. (2011) and Bulteau et al. (2015) for the GEV and GP distributions respectively. Both formulations require a perception threshold ($X_0$), above which the sample of historical information must be exhaustive. The key difference between the two formulations relates to the handling of sampling frequencies. Whereas Payrastre et al. (2011) deal entirely with AMAX data, Bulteau et al. (2015) adapt the methodology for POT observations and must therefore deal with an unknown number of censored events. In general, the likelihood function for the historical information is:

$$\mathcal{L}(Y \mid \theta) = \underbrace{\prod_{j=1}^{h_1} f_\theta(y_j)}_{(a)} \cdot \underbrace{\prod_{k=1}^{h_2} F_\theta(y_k^{ub}) - F_\theta(y_k^{lb})}_{(b)} \cdot \underbrace{(1 - F_\theta(X_0))^{h_3}}_{(c)} \cdot \underbrace{P(h \mid \theta)}_{(d)}, \tag{8}$$

where $f_\theta$ and $F_\theta$ is the probability density and cumulative distribution functions respectively. The terms in Eq. 8 give general expressions for different types of historical observations in $Y$, where $Y$ consists of $Y = h_1 + h_2 + h_3$ events. These terms describe (a) $h_1$ events with precise measurements, (b) $h_2$ events with upper ($y^{ub}$) and lower bounds ($y^{lb}$), and (c) $h_3$ events known to have exceeded the perception threshold but with no known upper limit. The term in (d) relates to the probability of observing h events exceed $X_0$ for the period of historical observation ($ny$) in years. For the GEV distribution and AMAX data, where there are a known number of missing observations, this term becomes:

$$P(h \mid \theta) = F_\theta(X_0)^{(n_y - h)}. \tag{9}$$

For the POT data and GP distribution, where observations have been sampled at an average frequency of $\lambda$ events per year, the number of exceedances of $X_0$ follow a Poisson process:

$$P(h \mid \theta) = \frac{(\lambda n_y [1 - F_\theta(X_0)])^h}{h!} \exp(-\lambda n_y [1 - F_\theta(X_0)]). \tag{10}$$

### 3.2 Markov chain Monte Carlo

Sampling of the posterior distribution $f(\theta \mid X, Y)$ is done using the Metropolis-Hastings MCMC algorithm (Metropolis et al., 1953; Hastings, 1970). Starting with an arbitrary parameter vector $\theta$, an iterative process is conducted where candidate vectors $\theta'$ are tested and either accepted or rejected based on the ratio of likelihoods:

$$\alpha = \frac{\mathcal{L}(X, Y \mid \theta')}{\mathcal{L}(X, Y \mid \theta_t)}, \tag{11}$$

where $\theta_t$ is the last accepted parameter vector. For a uniform random number $r \in [0, 1]$, the candidate vector $\theta'$ is accepted if $\alpha \geq r$, and thus $\theta_{t+1} = \theta'$, otherwise it is rejected and $\theta_{t+1} = \theta_t$. Candidate vectors are chosen based on a Gaussian distribution centred on the last accepted vector. The scale of this distribution is controlled so that the acceptance rate is around 25%.

The output of the MCMC is a large set of parameter vectors $\theta$ with densities $f(\theta|X, Y)$. For each vector, an extreme value model is defined, and quantiles of ESLs can be computed. Maximum likelihood estimates are associated with the mode of the set of vectors, and credibility intervals may be calculated based on the quantiles computed from the whole set of vectors. Naturally, the mode of the parameter vectors is that which maximises the likelihood function.

### 3.3 Data preparation, sampling and simulations

Before the Bayesian MCMC simulations can be performed, the sea level data must first be prepared and sampled. Data preparation involves removing long-term trends such as changes in MSL and ensuring observed events are measured relative to the standard vertical datum used in the region, normalhöhennull (NHN). For the AMAX data, this is done using monthly MSL values taken from the Permanent Service for Mean Sea Level (PSMSL: https://psmsl.org). Unfortunately, these records do not begin at the inception of the AMAX measurements (30 years short at Travemünde, 22 years at Wismar and 29 years at Warnemünde), instead we extend the PSMSL trends linearly to cover this shortfall. This method may not be the most appropriate due to accelerations in the rate of sea level rise. Although a quadratic trend results in differences of less than 1 cm at Travemünde and Warnemünde, at Wismar a maximum difference of approximately 5 cm is found. The use of a linear trend over a quadratic trend results in an increase to the AMAX samples not covered by the PSMSL data, which in turn leads to a positive bias of the final ESL estimates. However, it is unclear whether the quadratic trend would be better suited to the data, and in combination with the minor differences seen at Travemünde and Warnemünde, a linear trend is considered suitable for our purposes.

For the high-resolution tide-gauge data, MSL is calculated as a 1-year moving average of sea levels as suggested by Arns et al. (2013). Thankfully, most historical measurements are recorded relative to MSL at the time, and thus no correction is needed. Historical events measured to NormalNull (NN), the old standard vertical datum used in Germany, are detrended by transferring the measurements to NHN using local adjustment values and removing MSL using the available PSMSL data.

Given the availability of AMAX and high-resolution data, we employ two approaches to sample ESLs which are described in Section 2.2. First, AMAX sampling is an obvious choice as long records of AMAX data are available at Travemünde, Wismar and Warnemünde. These records provide the largest water levels for each hydrological year, starting on the 1st of November and ending on the 31st of October the following year. While no further sampling is required, each of these records ends in 2009, 11 years before the end of the high-resolution tide-gauge data. To maximise the available data for our analyses, each

AMAX set is extended by sampling from the tide-gauge data over the missing years (2009-2020). As a result, the AMAX data consists of 195 ESLs at Travemünde, 155 at Wismar and 147 at Warnemünde. For the high-resolution data, ESLs are sampled using the POT method. We follow the approach outlined by MacPherson et al. (2019) for threshold selection, who conducted an analysis of EVA techniques for the German Baltic coast. Here, a threshold equal to the 98th percentile of high-water peaks is chosen. Before selecting the threshold, peak water levels are declustered using an interval of 3 days to ensure independence.

All ESLs that exceed this threshold are extracted from the record to form the ESL sample.

       As mentioned in Section 3.1, it is a necessary condition that the available historical information is exhaustive above a perception threshold. That is, the only events which have exceeded the perception threshold for the duration of historical observation exist within the historical record. Therefore, the perception threshold should be set high enough to ensure this assumption is true. At first, a systematic approach to setting a perception threshold was attempted based on the systematic data

and the period of historical information. Here, ESLs were estimated using systematic data only for return periods dependent on the number of historical events available and the length of the historical record. For example, the perception threshold might be set to a height equivalent to a 1-in-100 year event, where a 200 year long historical record is available which contains 2 events. However, due to large differences in the magnitude of systematic and historical observations, relying on the systematic data alone was not sufficient, and no one method could be applied at all sites. Instead, perception thresholds were chosen on a

site-by-site basis, using all available data for each case.

       Given the lack of a clear physical threshold at any of the tested locations (e.g. a sea wall where all exceeding events are recorded), a threshold selection process was conducted based simply on the author's intuitive reasoning. Factors that influenced the selection process include the magnitude and occurrence of ESLs in both the systematic and historical records and the length of the historical record in question. Keeping in mind the assumption that the historical record is exhaustive, and due

to the subjective nature of this method, final perception thresholds were set conservatively high at 2.3 m at Flensburg, 2 m at Schleswig, 2.25 m at Eckernförde, 2.25m at Kiel, 2.6 m at Travemünde, 2.25 m at Wismar and 2 m at Warnemünde. Historical ESLs that do not exceed the perception threshold cannot be used in the analysis, and are thus disregarded. These events are highlighted in Figure 2.

       ESLs at each site were modelled using the Bayesian MCMC approach described in Section 3.1 and 3.2. Although traditional

EVA may be performed using the systematic data alone, the Bayesian MCMC approach is used in all cases for comparison purposes. We use the GP distribution to model POT data (POT-GP) and the GEV distribution to model AMAX data (AMAX-GEV). Depending on data availability at each site, we perform four separate analyses using: 1) POT samples only, 2) POT samples with historical measurements, 3) AMAX samples only, and 4) AMAX samples with historical measurements. At Wismar and Warnemünde, as no historical records are available, ESLs from the AMAX records are used in lieu of historical

measurements for the second analysis. For the POT-GP analyses, the sampling frequency ($\lambda$) and location parameter ($\mu$) are assumed to be constant for the combined period of historical and systematic observation. At each site, $\mu$ is equal to the threshold used to sample the ESLs. Hence, only the shape ($\xi$) and scale parameters ($\sigma$) of the GP distribution are considered. In contrast, all three parameters of the GEV distribution ($\xi$, $\sigma$, $\mu$) are sampled.

### 3.4 ESL stationarity, long-range dependence and variability

From the AMAX record at Travemünde (Figure 3), a decrease in the number of large ESL events over the past century can be seen. Of these events (>95th percentile), 7 occurred prior to 1920 with only 3 occurring afterwards. We perform several analyses to assess the stationarity and long-range dependence of ESLs at Travemünde and to identify periods of low and high ESL activity.

     Stationarity refers to the property of a stochastic process or time series where the statistical properties of the process remain

constant over time. More specifically, a stationary process has a constant mean, constant variance, and constant autocovariance over time. As the methods used to estimate ESLs in this study assume the underlying data is stationary, we first perform KPSS (Kwiatkowski et al., 1992) tests on samples of ESLs at all sites for both POT and AMAX samples to confirm this. The KPSS test evaluates the cumulative sum of the deviations from the estimated trend in the time series. If this sum exceeds some critical value based on the sample size, the null hypothesis of stationarity is rejected, indicating the presence of a unit root and non-

stationarity. On the other hand, if the sum is below the critical value, the null hypothesis cannot be rejected, suggesting the series is stationary.

     Next, we check for long-range dependence within the AMAX record which describes the persistence or correlation between distant observations. We quantify long-range dependence using the Hurst exponent (Hurst, 1951), which is a measure of the strength of the correlation between observations that are far apart in time. For a series of N observations, the Hurst exponent

is calculated using the rescaled range (R/S) analysis. This involves creating a mean adjusted series and segmenting the observations into smaller subsets of size n < N. For each segment, the range (R) between the maximum and minimum values is calculated. The rescaled range (S) is then determined by averaging the range of all segments and dividing by the standard deviation of the entire series. This is repeated for a number of segment sizes to account for both short and long periods of observations. Given the 195 observations available at Travemünde, we chose to use segment sizes of n = 3, 6, 12, 24, 48, 96.

The Hurst exponent is found by plotting the log of the rescaled range (log(S)) against the log of the segment size (log(n)) and fitting a straight line. The Hurst exponent is thus the slope of the fitted line. A Hurst exponent greater than 0.5 indicates long-range dependence, while a value less than 0.5 indicates short-range dependence or anti-persistence.

     To identify periods of low or high ESL activity, EVA is applied using an iterative process where AMAX data is limited to those events occurring within a 70-year moving window. A window size of 70 years was chosen to match the length of the high-

resolution tide-gauge record at Travemünde (71 years in length). For each iteration, the Bayesian MCMC is applied, and the window is then moved forward one year. Maximum likelihood estimates of HW200 and their corresponding 95% credibility intervals are computed at each step from the resulting set of GEV parameter vectors. The analysis was performed twice,

once with systematic data only and again with historical information included. For the second case, all available historical information is used, even those events which occur outside of the 70-year window.

Lastly, we consider how the sample size of the systematic data affects ESL estimates. Here, probability density estimates of HW200 were computed for samples of AMAX observations ranging in size from 70 to 195 events. At each tested sample size, 10,000 sets of AMAX observations were generated using bootstrap sampling, from which an equal number of HW200 maximum likelihood estimates were made. Probability density estimates of HW200 at each sample size are calculated for comparison.

## 4 Results

### 4.1 ESL estimates

Incorporating historical information in the analysis of ESLs along the German Baltic coast results in significant changes to ESL estimates. Estimates of HW200 and HW1000 (1-in-1000 year ESL event) at all tested locations are given in Table 2, including the upper and lower bounds of the 95% credibility intervals. Maximum likelihood estimates from the POT-GP analyses increase at all sites when historical information is included, by as little as 16.5% (33 cm) at Warnemünde and as much as 47.9% (75 cm) at Schleswig. There are decreases in the range of the 95% credibility intervals at four of the seven tested locations. This is most evident at Travemünde where a decrease of 51.4% occurs at the HW200 level. At sites where AMAX data was considered, the effect of incorporating historical information can only be examined at Travemünde. Here, changes in the maximum likelihood estimation are negligible (< 1%) at both HW200 and HW1000. However, uncertainties in the form of 95% credibility intervals decrease by approximately 42%, which is equivalent to 28 cm at HW200 and 48cm at HW1000.

To illustrate the benefits of incorporating historical information in EVA, Figure 4 shows the results of the MCMC method at Travemünde for both POT-GP and AMAX-GEV analyses. While the high-resolution tide-gauge record is not considered short and is in fact the longest available record of its type along the German Baltic coast, there are large differences between estimates of ESLs made using the POT sample taken from the tide-gauge record and the much longer AMAX data set (71 vs. 195 years). However, by incorporating historical information in EVA, these discrepancies are substantially reduced. For example, maximum likelihood estimates of HW200 and HW1000 differ by 45 cm and 64 cm respectively between the POT-GP and AMAX-GEV analyses. These values are reduced to 5 cm at HW200 and 19 cm at HW1000 when historical information is included.

While changes to the maximum likelihood estimates in the POT analysis is stark, perhaps more apparent is the reduction in uncertainties. For instance, the range of estimates within the 95% credibility intervals decreases by approximately 54% and 42% at HW200 for the POT-GP and AMAX-GEV analyses respectively, when historical information is included. Similar values can also be seen at HW1000. Interestingly, most of the reduction in uncertainty occurs at the lower bound for the POT-GP analysis and the upper bound for the AMAX-GEV analysis.

Similar benefits can also be seen at sites other than Travemünde. Figure 5 shows maximum likelihood estimates and 95% credibility intervals for HW200 at all sites considered in this study. Although uncertainties increase at Flensburg, Schleswig and

**Table 2.** Estimates of HW200 and HW1000 at all sites (rounded to the nearest cm). The first figure provides the maximum likelihood estimate while the range of the 95% credibility intervals are given in square parentheses with lower and upper bounds in round parentheses below. Changes to maximum likelihood estimates are given as percentages, with changes to the range of the 95% uncertainty bounds are shown in square parentheses.

| | | HW200 | | | HW1000 | | |
|---|---|---|---|---|---|---|---|
| | Site | Syst. Only | Syst. + H.I. | Change | Syst. Only | Syst. + H.I. | Change |
| **POT-GP** | Flensburg | 1.95 [0.60] | 2.49 [0.66] | 27.9% | 2.04 [0.93] | 2.93 [1.20] | 43.3% |
| | | (1.84-2.44) | (2.28-2.93) | [9.8%] | (1.90-2.83) | (2.57-3.77) | [28.7%] |
| | Schleswig | 1.56 [0.67] | 2.30 [0.78] | 47.9% | 1.59 [1.01] | 2.86 [1.60] | 79.7% |
| | | (1.51-2.18) | (2.05-2.84) | [16.6%] | (1.53-2.54) | (2.42-4.02) | [58.4%] |
| | Eckernförde | 2.07 [1.77] | 2.52 [1.19] | 21.7% | 2.24 [3.23] | 3.02 [2.35] | 35.1% |
| | | (1.90-3.67) | (2.22-3.40) | [-33.0%] | (2.00-5.23) | (2.54-4.89) | [-27.3%] |
| | Kiel | 2.07 [1.67] | 2.50 [1.13] | 20.4% | 2.25 [3.02] | 2.98 [2.19] | 32.7% |
| | | (1.90-3.57) | (2.21-3.34) | [-32.4%] | (2.00-5.02) | (2.52-4.71) | [-27.6%] |
| | Travemünde | 2.22 [1.14] | 2.72 [0.55] | 22.7% | 2.43 [2.01] | 3.29 [1.08] | 35.4% |
| | | (2.03-3.17) | (2.52-3.07) | [-51.4%] | (2.15-4.16) | (2.92-4.01) | [-46.0%] |
| | Wismar | 2.21 [0.97] | 2.66 [0.98] | 20.5% | 2.40 [1.64] | 3.14 [1.93] | 30.8% |
| | | (2.02-2.99) | (2.39-3.37) | [1.3%] | (2.13-3.77) | (2.69-4.62) | [17.2%] |
| | Warnemünde | 1.99 [0.95] | 2.32 [0.85] | 16.5% | 2.20 [1.72] | 2.77 [1.64] | 25.9% |
| | | (1.82-2.77) | (2.08-2.93) | [-10.9%] | (1.94-3.66) | (2.37-4.01) | [-4.5%] |
| **AMAX-GEV** | Travemünde | 2.67 [0.66] | 2.68 [0.38] | 0.6% | 3.07 [1.10] | 3.10 [0.63] | 0.8% |
| | | (2.47-3.13) | (2.53-2.92) | [-42.1%] | (2.77-3.86) | (2.86-3.49) | [-42.7%] |
| | Wismar | 2.68 [0.74] | - | - | 3.07 [1.24] | - | - |
| | | (2.46-3.20) | - | - | (2.72-3.96) | - | - |
| | Warnemünde | 2.27 [0.66] | - | - | 2.62 [1.09] | - | - |
| | | (2.10-2.76) | - | - | (2.35-3.45) | - | - |

Wismar, these also coincide with large increases to the maximum likelihood estimates. Large reductions in uncertainties can be seen at Eckernförde and Kiel, with a slight reduction at Wismar. Where only systematic data is considered, a large proportion of uncertainty lies above the maximum likelihood estimates (Flensburg: 82.2%, Schleswig: 92.8%, Eckernförde: 90.5%, Kiel: 89.5%, Travemünde: 83.5%, Wismar: 80.6% and Warnemünde: 82.3%). Understandably, the inclusion of large historical events

reduces uncertainty in the upper bound of the credibility intervals, proportionally if not quantitatively (Flensburg: 67.6%, Schleswig: 68.3%, Eckernförde: 74.7%, Kiel: 74.6%, Travemünde: 62.6%, Wismar: 72.7% and Warnemünde: 72.3%).

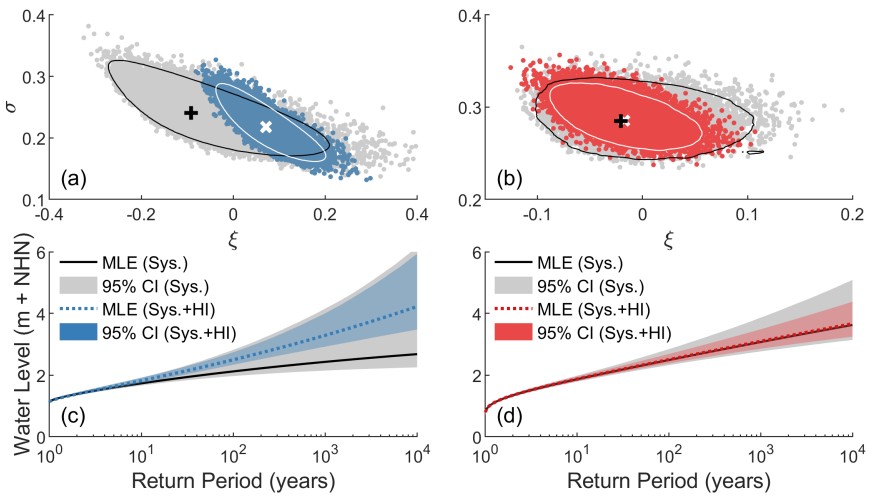

**Figure 4.** Results of the MCMC at Travemünde (blue: POT-GP, red: AMAX-GEV): **(a)** and **(b)** scatter plots of the shape ($\xi$) and scale ($\sigma$) parameters. For analyses conducted using systematic data only, sampled parameter pairs are shown as grey dots with the mode of the distribution shown as a black plus. Analyses conducted using both systematic data and historical information show sampled parameter pairs as coloured dots with the mode of the distribution shown as a white cross. 95th percentile contour lines are shown as black for analyses of systematic data only and white for systematic data and historical information. Return water level plots are shown in **(c)** and **(d)** for the POT-GP and AMAX-GEV analyses respectively, including maximum likelihood estimates (MLE) and 95% credibility intervals (CI).

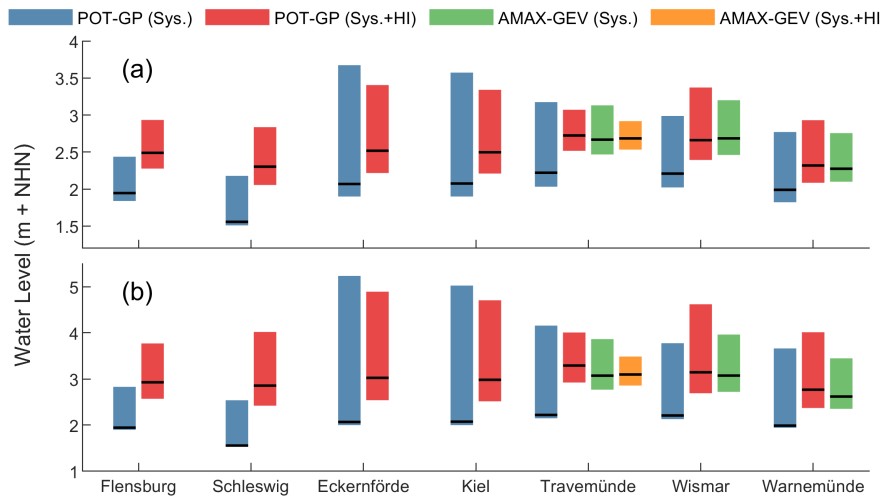

**Figure 5.** Comparison of **(a)** HW200 and **(b)** HW1000 estimates at all sites. Maximum likelihood estimates are shown as black horizontal lines. 95% credibility intervals are shown as colored bars. Where historical information is included, ESL estimates increase at all sites and credibility intervals are generally reduced. This occurs for both the POT-GP and AMAX-GEV analyses.

**Table 3.** Return period estimates (years) of the 1872 event at all sites. Estimates that are 'Undefined' represent values which lie outside the range of the maximum likelihood distribution. Values that could not be found due to insufficient data are shown as dashes (-).

| Site | POT-GP (Syst.Only) | POT-GP (Syst. + H.I.) | AMAX-GEV (Syst.Only) | AMAX-GEV (Syst. + H.I.) |
|---|---|---|---|---|
| **Flensburg** | Undefined | $1.57 \times 10^3$ | - | - |
| **Schleswig** | Undefined | $2.86 \times 10^3$ | - | - |
| **Eckernförde** | Undefined | $1.63 \times 10^3$ | - | - |
| **Kiel** | Undefined | $1.37 \times 10^3$ | - | - |
| **Travemünde** | $5.59 \times 10^{11}$ | $1.90 \times 10^3$ | $6.80 \times 10^3$ | $5.82 \times 10^3$ |
| **Wismar** | $3.99 \times 10^6$ | $0.73 \times 10^3$ | $0.90 \times 10^3$ | - |
| **Warnemünde** | $4.23 \times 10^6$ | $2.29 \times 10^3$ | $6.76 \times 10^3$ | - |

Looking only at the sites where long AMAX records are available (Travemünde, Wismar and Warnemünde), we see much better agreement between the POT-GP analyses including historical information and the AMAX-GEV analyses of systematic data only, despite the significantly shorter tide-gauge records. Differences in HW200 estimates decrease from 45cm, 48 cm and 28 cm at Travemünde, Wismar and Warnemünde respectively, to 5 cm, 2 cm and 5 cm.

Including historical information also allows for a more reasonable representation of the 1872 event. Table 3 shows the estimated return period in years of the 1872 event at each site and for each analysis. Given only high-resolution tide-gauge data, return period estimates of 1872 are not realistic, suggesting that the event is an outlier. At travemünde, 1872 is estimated to have a return period of more than 500 billion years. Furthermore, no estimates could be made at Flensburg, Schleswig, Eckernförde or Kiel, as the 1872 value is not defined within the resulting maximum likelihood distributions. At Wismar and Warnemünde, estimated return periods are also high at approximately 4 million years. When historical information is included, return periods of between 700 and 2860 years are assigned to the 1872 event. These estimates are in the same order of magnitude provided by the AMAX-GEV analyses, which include the 1872 event within the systematic data.

## 4.2 ESL variability

Before considering ESL variability, stationarity was confirmed at each site using the KPSS test for POT samples and AMAX samples where available. In regards to the AMAX data at Travemünde, we find that the series exhibits strong long-range dependence with a Hurst exponent of 0.69. Similarly high Hurst exponents were found at Wismar (0.77) and Warnemünde (0.62). This suggests that there is persistency in the series of ESLs at Travemünde, Wismar and Warnemünde which can be seen as some long-term variability. This is likely to result in fluctuations in the occurrence of ESLs with periods of many large events and periods of many small events.

The discovery of long-range dependence in the series of AMAX ESLs at Travemünde in combination with large increases in ESL estimates due to the inclusion of historical information raises questions about the stability of estimates made using

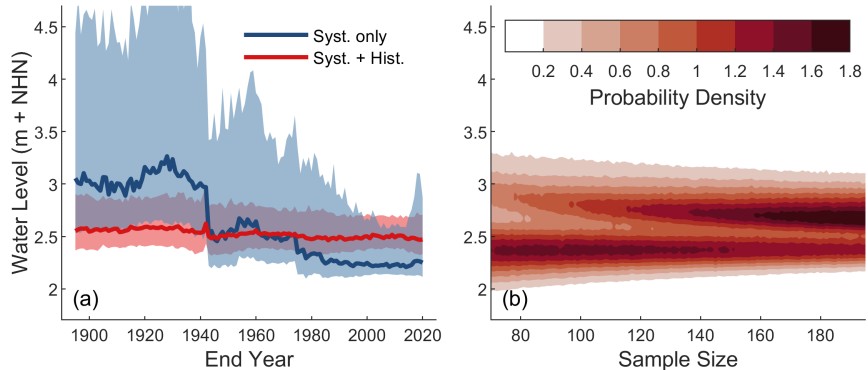

**Figure 6. (a)** Maximum likelihood estimates and 95% credibility intervals of HW200 at Travemünde, estimated using a 70 year moving window of AMAX values, both omitting (blue) and including (red) historical information. **(b)** Probability density estimates of HW200 made using bootstrap samples of AMAX data of different sizes.

only short tide-gauge records. As described in Section 3.4, we assess changes in HW200 estimates over time by (a) performing traditional AMAX analyses based on a moving window of 70 years and (b) comparing probability density estimates from bootstrap AMAX samples of different sizes. For both analyses, AMAX observations from Travemünde were used. Results are shown in Figure 6.

Considering only the systematic data in analysis (a), there is a clear downward trend in the maximum likelihood estimates of HW200, and general decrease in the range of 95% credibility intervals. The largest change occurs in 1943, when the AMAX window no longer includes the exceptionally large 1872 event. The largest HW200 estimate of 3.21 m occurs for the period of 1858 to 1927, with the smallest estimate of 2.22 m occurring for the period of 1943 to 2012, a difference of approximately 1 m. In contrast, HW200 estimates made using systematic data in combination with historical information differ by a maximum of only 14 cm. The large variability of HW200 estimates made using systematic data alone, and by contrast the generally stable estimates made with historical information included, is also reflected in the range of 95% credibility intervals. Before 1943, the difference between upper and lower bounds of the 95% credibility intervals range from 1.70 m in 1907 to 2.63 m in 1929, for estimates made using systematic data only. Afterwards, this range generally decreases to a minimum of 0.49 m in 2008. For estimates made using both systematic data and historical information, these ranges are more stable, decreasing somewhat steadily from a maximum of 0.58 m in 1896 to 0.32 m in 2010.

For the second analysis (b), the resulting probability density contours of the HW200 estimates (Figure 4b) show two clear regions of high probability density. The lower region exists at a HW200 value of approximately 2.4 m and is dominant for sample sizes < 140. At higher sample sizes, the upper region becomes dominant and corresponds to the HW200 estimate of 2.66m derived using the full systematic AMAX record. The upper region of high probability is clearly a result of the presence of the exceptionally large 1872 event in the data. When the 1872 event is removed from the analysis (results not shown), the upper region disappears entirely, and the probability density of the lower region grows substantially.

## 5 Discussion

Estimates of current ESLs can be dramatically improved by incorporating historical information in EVA. For the German Baltic coast, estimates made using limited data from high resolution tide-gauges in combination with historical information show good agreement with estimates made using much longer AMAX records. Furthermore, we show that even long records of systematic data can benefit from the inclusion of historical information in terms of reduced estimate uncertainties. These results support the conclusions of several studies which consider the benefits of including historical information in EVA (Bulteau et al.,

2015; Payrastre et al., 2011; Benito et al., 2004; Hamdi et al., 2015; Coles and Tawn, 2005; Reis and Stedinger, 2005; Coles et al., 2003).

     At all sites tested along the German Baltic coast, combining historical information with tide-gauge records results in large increases to the estimates of ESLs. Of the seven sites tested, four benefit in terms of reduced estimate uncertainties. At all sites, including those where uncertainties increase, the proportion of uncertainty in the upper bound decreases. This is to be expected

given that historical measurements provide valuable information in the upper range.

     The results presented in this study raise concerns regarding the estimates of ESLs made using limited data. It is generally accepted that estimative return periods should not exceed four times the length of the record considered (Pugh, 2004), in order to maintain manageable uncertainties. Despite this, we show that even long records can lead to severely underestimated ESLs despite fulfilling this criterion. Indeed, at most sites tested in this study, incorporating historical information results in HW200

estimates that remain within the upper uncertainty bounds of the initial estimates, highlighting the importance of quantifying uncertainties in the first place. However, at Flensburg and Schleswig, updated HW200 estimates exceed the 95% credibility intervals computed using systematic data only.

     For the case of Schleswig, comparisons between events recorded at the two nearest sites of Flensburg and Eckernförde show that within the tide-gauge data, ESL events are typically much smaller. However, these large differences are not reflected in

the historical record. Changes in the local bathymetry, including development of the sand spit at the mouth of the Schlei inlet, likely explains the reduced size of ESLs at Schleswig between the two records. Consequently, the use of historical information in EVA at this site would not be applicable, as the systematic and historical ESLs would be drawn from different distributions. ESL estimates provided in this paper for Schleswig should thus be approached with caution, and not considered accurate or reliable.

Like Schleswig, Flensburg lies at the end of an inlet connected to the Baltic Sea, however the Flensburg Firth is much larger than the narrow Schlei. Given the larger size and lack of development at the mouth of the Flensburg Firth, it is unlikely that changes in the local bathymetry would result in significant changes in the attenuation of ESLs. Instead, it appears that the available systematic data is simply insufficient for the estimation of large ESLs, such as HW200. Given the tide-gauge record at Flensburg is one of the longest available along the German Baltic coast, spanning 66 years, and ESLs in the region are

influenced by the same large-scale atmospheric forcing, we must question the validity of HW200 estimates along the entire coastline that rely solely on tide-gauge data. This finding is supported by the results at Travemünde, where large changes occur due to the inclusion of historical information, despite a relatively long tide-gauge record.

Long-range dependence in the series of AMAX ESLs at Travemünde may explain the underestimation of ESLs when only limited tide-gauge data is used. The discovery of persistence in the data suggests the presence of some long-term variability. While the cause of this variability is not known and considered outside the scope of this paper, it would suggest that available tide-gauge records along the German Baltic coast cover a period of relatively low ESL activity. In such a case, direct EVA methods which rely on high-resolution tide-gauge data, in addition to JPM and RFA, would be prone to underestimations due to bias within the available data.

Based on the extensive AMAX record at Travemünde, a period of high ESL activity took place prior to 1927 and has since been followed by a period of lower extremes. Further research is necessary to establish if this trend is the result of permanent changes in the mechanisms driving ESLs in the region, or simply some long-term variability not yet defined. However, research conducted by Jensen and Töppe (1990) and Jensen et al. (2022), in combination with the high Hurst exponent found in this study suggest the latter. The occurrence of large events before the introduction of systematic records, not equal in height to 1872 but of comparable magnitude, argue against the notion that the decrease in ESL activity is permanent. Furthermore, we show that the inclusion of historical information in EVA leads to stable estimates of ESLs regardless of the systematic period analysed. Thus, we believe that some long-term variability in ESLs along the German Baltic coast is responsible for the apparent low ESL activity over the past century. If the current phase of low ESL activity is indeed the result of long-term variability, this has significant implications for the management of coastal areas along the German Baltic coast given the lack of long sea level records. In such a case, the method described herein is a valuable tool where historical information is available.

The largest influence on the results of this study is the event of November 1872, which caused exceptionally high water levels along much of the German Baltic coast. Differences in estimates produced using systematic tide-gauge data and AMAX records are largely due to this event. In fact, 1872 was the only event considered in combination with tide-gauge data at three of the seven sites tested, however changes to the maximum likelihood estimates due to its inclusion are not indifferent to results at other sites where multiple historical events are considered. Due to the exceptional magnitude of the event, it has until recently been treated as an outlier and thus excluded from statistical analyses. Hofstede and Hamann (2022) argue that based on the series of AMAX ESLs at Travemünde, 1872 is indeed an outlier given that it is more than 50% higher than the second largest event. Indeed, Mudersbach and Jensen (2009) assessed the return period of a corrected 1872 event at about 10,000 years. However, they concluded that the event could not be well defined statistically given the limited sample population, and suggested extending the available data using historical information. Jensen et al. (2022) highlight the occurrence of events within historical records of similar magnitudes to 1872. Given these events, Jensen et al. (2022) argue that 1872 should not be considered an outlier and that the systematic records are not sufficiently long to deal with events of these magnitudes.

When considering the full historical record at Travemünde in combination with high-resolution tide-gauge data, we assess the return period of the 1872 event to be approximately 1,900 years. Combining the long AMAX record with historical information provides a return period estimate of approximately 5,800 years. Given the length of the historical record at Travemünde (approximately 980 years), and the occurrence of other large ESLs within it (1320, 1625, 1694), we agree with the arguments of Jensen et al. (2022) that the 1872 event should not be considered an outlier, but rather an exceptional realisation of the underlying ESL distribution, and we would recommend for its use in EVA. While sea level records that cover a period that includes

1872 can provide very good ESL estimates using traditional EVA methods, only few sufficiently long records exist along the German Baltic coast (including the AMAX records described herein). Despite this, we show that even short tide-gauge records (approximately 30 years in our case) with one measurement of 1872 can provide similar results. Therefore, reconstructions of past extremes offer valuable information with which to improve EVA.

Large differences exist in the estimates of ESLs made using either the POT-GP or AMAX-GEV approaches. While the incorporation of historical information reduces these differences, it does not provide any insight into which method performs best. Indeed, the POT-GP approach is generally preferred in the literature (Arns et al., 2013; Wahl et al., 2017), but this does not necessarily apply to the case of the German Baltic Sea coast. We find that when both methods are constrained to the same record length (see Appendix A), the POT-GP method generally performs better with lower uncertainties at the distribution tails. At all sites, the AMAX-GEV provides larger ESL estimates at high return periods. Interestingly, including historical information in the analysis produces a different result, with the AMAX-GEV analysis providing lower uncertainties at high return periods. One possible explanation for this involves the sampling threshold for the POT method. We assume that this threshold is constant for the full duration of historical and systematic observation, following the study by Bulteau et al. (2015), but this may not be the case. Indeed, large differences in results due to the inclusion of historical information suggests this assumption may be false. Thus, an advantage of the AMAX-GEV approach is that no sampling threshold is required. Given a single sea level record with no historical information, we would recommend the POT-GP approach over the AMAX-GEV due to the reduced estimate uncertainties. However, the AMAX-GEV approach may provide more precise results when historical information is available. Where a longer AMAX record is available, such as in this study, the AMAX-GEV approach provides clearly better results due to the increased data.

We confirm that exceptionally large events commonly considered to be outliers can be placed within classical EVA using Bayesian techniques (Bulteau et al., 2015; Coles and Tawn, 2005; Reis and Stedinger, 2005), and demonstrate this for the case of the 1872 event along the German Baltic coast. Much discussion surrounding the suitability of 1872 and other supposed outliers in statistical analyses exists (Jensen et al., 2022; Hofstede and Hamann, 2022; MacPherson et al., 2019; Hamdi et al., 2015). While scientific and coastal management perspectives on its importance may differ, arguments tend to revolve around the consequences of such an event occurring or not. That is, what are the costs on one hand to implement coastal defences for an event that might not occur again, and on the other hand, what are the potential damages. Due to the exceptional height of the 1872 ESL event, we agree with (Jensen et al., 2022) that design water levels do not necessarily have to be based on the largest past event, but all available relevant information should be used for their derivation.

There exists a gap between the design heights of coastal defences and what is actually possible. The German Baltic coast, and Travemünde in particular, is a good example of this fact. The 1872 event produced exceptionally high water levels as a result of a rare sequence of atmospheric forcing. Design water levels in the region, based on a return period of 200 years, are thus much lower than what was observed during this event. Even after including the 1872 measurements, estimates of HW200 remain well below the level of the 1872 ESL. Although this can be addressed by raising design heights to that of the largest event on record, this would require significantly large investments which may be of better use elsewhere. Furthermore, this strategy of coastal defence planning is reactive rather than proactive. Instead, integrated flood risk management should be implemented

with an aim to reduce vulnerability to coastal floods via appropriate urban planning based on current and future ESLs, and investment into advanced warning systems, evacuation planning and emergency response networks in exposed regions.

## 6  Conclusions

In this study, we combine systematic tide-gauge data with historical information using a Bayesian MCMC method to assess ESLs at seven sites along the German Baltic coast. In general, we find the inclusion of historical information in EVA is beneficial, resulting in reduced estimate uncertainties and the incorporation of exceptionally large events that would otherwise be considered outliers.

At the German Baltic coast, the incorporation of historical information in EVA results in large increases in the estimates of ESLs. At all sites, even those with long systematic records, (Flensburg: 66 years, Travemünde: 71 years, Wismar: 63 years and Warnemünde: 67 years), estimates for HW200 increase by between 16-28%. We find the presence of long-range dependence in the series of ESLs at Travemünde, which suggests that long-term variability is affecting extremes in the region. Further, we show that recent ESL activity has been relatively low. As this period of low ESL activity covers the full period of systematic observation, we caution against the use of systematic data alone when assessing ESLs in the region.

Estimates of ESLs in the region are largely affected by an exceptional event that took place in 1872. Although it is commonly dismissed as a statistical outlier, we argue against this notion and advocate for its inclusion, partly due to its substantial impact on estimates of ESLs. Where available, incorporating the 1872 event into analyses of ESLs based on limited systematic data provides similar results to analyses which employ much longer sea level records. Sourcing further information on this event may allow for improved ESL estimates at other sites where only limited systematic data is available. Such information may be found in the field, by consulting historical records or by conducting re-analyses of the event using numerical models.

*Data availability.* A number of sea level data were used in this study. Systematic tide-gauge data was sourced from the GESLA-3 database (Haigh et al., 2022) in combination with data from MacPherson et al. (2019), provided by Kelln et al. (2017). The latter is available from local water and shipping authorities along the German Baltic Sea coast upon request. Annual maxima sea levels at Travemünde, Wismar and Warnemünde were supplied by the Ministry of Agriculture, Environment and Consumer Protection Mecklenburg-Western Pomerania upon request. Monthly mean sea level data was downloaded from the Permanent Service for Mean Sea Level (https://psmsl.org/). Water levels of historical events were sourced from literature (Jensen and Töppe, 1990; Jensen et al., 2022).

## Appendix A:  ESL Sampling

In this study, we employ two different approaches for sampling ESLs. The first technique, POT is generally preferred in literature for reasons explained in Section 2.2. Wahl et al. (2017) also note that AMAX sampling may result in larger uncertainties at the tails of the distribution when sea level records are short. While we are constrained by the records sourced from MLUV (2012), which are only available as AMAX samples, this is not the case for the high-resolution tide-gauge data. Thus, we

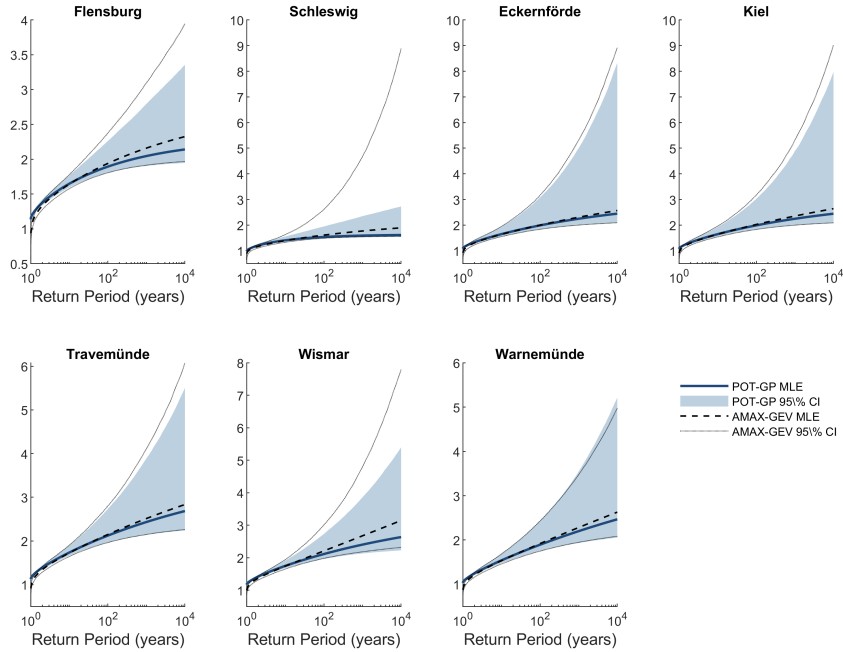

**Figure A1.** Comparison of POT-GP and AMAX-GEV approaches to the Bayesian MCMC method for estimating ESLs. At each site, estimates of ESLs are made using high-resolution tide-gauge data only. In general, the AMAX-GEV approach results in higher estimates of ESLs at high return periods and larger uncertainties at the tails of the distributions.

decided to employ POT sampling for these records due to their preference in literature, and so that we may demonstrate the use of the Bayesian MCMC EVA method for both POT-GP and AMAX-GEV approaches. To directly compare the two approaches,
we also performed an AMAX-GEV analysis using the high-resolution tide-gauge data. Figure A1 shows ESL estimates including 95% credibility intervals estimated at each site using the high-resolution tide-gauge data only. At all sites, the AMAX-GEV method results in larger estimates of ESLs at high return periods. Also of note are the larger uncertainties at the tails of the distribution at all sites except for Warnemünde, which supports the findings of Wahl et al. (2017). In general, the POT-GP appears to produce more reliable results given the same record duration based purely on the reduced uncertainties.
Next, we considered how these estimates are affected by the addition of historical information. We performed the analysis again, but included historical information with measurement uncertainties given by Jensen et al. (2022). Results are shown in Figure A2. As with the first analysis, both the POT and AMAX samples are taken from the same high-resolution tide-gauge record. For both POT-GP and AMAX-GEV analyses, the introduction of historical information is beneficial in terms of reduced estimate uncertainties. Interestingly, we find that the AMAX-GEV approach performs better in terms of reduced uncertainties
at all sites except Schleswig. Differences in the maximum likelihood estimates between the two analyses are much reduced.

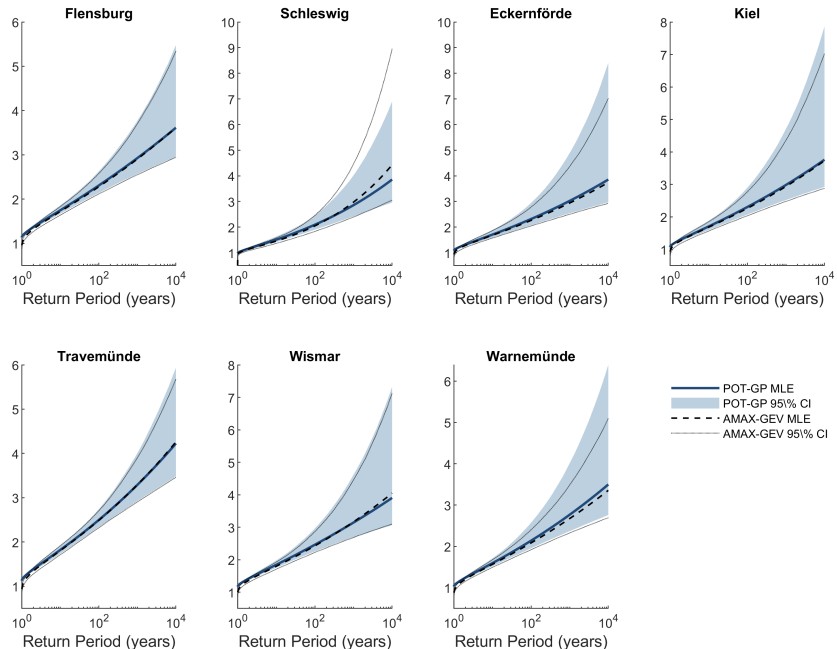

**Figure A2.** Comparison of POT-GP and AMAX-GEV approaches to the Bayesian MCMC method for estimating ESLs including historical information. At each site, estimates of ESLs are made using the same high-resolution tide-gauge record in combination with historical information. In contrast to results where historical information is omitted, the AMAX-GEV approach performs somewhat better than the POT-GP approach in terms of reduced uncertainties at the distribution tails. Differences in the maximum likelihood estimates between the two methods are much reduced.

*Author contributions.* This study was conceived and conceptualized by LM and AA. Data curation and formal analysis was performed by LM, as well as visualisation and writing. Development of the methodology was conducted by LM with insight provided by SF and AA. AA managed and administrated the funding acquisition for conducting the research and provided supervision and validation of the methodology and discussion points. Further validation was provided by FM and JJ.

*Competing interests.* There are no competing interests.

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
