# Peer review of "Bayesian extreme value analysis of extreme sea levels along the German Baltic coast using historical information"

_EGUsphere, 2023_

## Referee Comment (RC1)

The paper "Bayesian extreme value analysis of extreme sea levels along the German Baltic coast using historical information" discusses the inclusion of historic extreme events, often disregarded as outliers, into extreme value analysis and adapts an established approach to the German Baltic coast. This new methodology is able to reduce uncertainties at some of the locations of the study and to shows that current design approaches might underestimate extreme sea levels significantly due to limited availability of long tide gauge records. The paper further discusses large scale variability found in the Travemünde record, which remains undefined but might be the reason for the underestimation of extreme sea levels in other records.

The authors present their research in a precise and easy to understand manner. Their work highlights the next step in possible improvements to the coastal protection in their study area and given a well-founded recommendation on the treatment of outliers in EVA.

I therefor recommend the publication after some minor revisions, mainly focused on some clarifications and extended discussions of its important findings.

General comments:

*Data*: The length of most of the hourly records does not seem to match the length of the GELSA3 records. Please provide the data sources used to extend the records.

*Data preparation and filtering*: A few short comments on the methods of data preparation and their potential influence on the results seem helpful for the reader. Specifically, these questions arise:

- While certainly the easiest methods of detrending, extending the linear trend into the first half of the 19th century (line 168) seems prone to overestimate sea level rise in extended timespan due to possible acceleration between the 19th and 20th century. This requires at least a short discussion of the underlying assumptions and their uncertainties and whether other option to fill this gap (e.g. a quadratic trend) were tested.
- The authors explain sufficiently, why an additional threshold for the selection of the historic events is necessary, but find that no systematic method was able to derive an appropriate threshold for all tide gauges. They therefor resort to a manual review of data. To understand the process of this selection, a description of the (possibly subjective) criteria for this review would be helpful to the reader.
- Furthermore, the consequences of this selection remain unclear until much later in the paper, where it is briefly mentioned that only the 1872 event remains at four stations after applying this threshold (line 355-357). This is important information for the reader and should be mention alongside the definition of the filter.
- In this context, it also is unclear which step of data preparation Fig. 2 depicts. Some parts of the data preparation (the extension of the AMAX timeseries) seems to have already been conducted, others (applying the threshold for historical data) not.

*Results (ESL estimates)*: The results are presented in an overall clear and easy to understand fashion. Still, an addition to Fig. 4 would highlight the results of the study even more. I suggest to include the AMAX-GEV (Sys.+HI) estimate of Travemünde as well to show that the AMAX estimate benefits from the inclusion as well (even though the interpretation is limited by the availability of only one example). I further suggest adding a figure similar to Fig. 4, but depicting the HW1000 estimates, as supplementary material. These values are already given in Table 2, but I found Fig. 4 to enable a much easier comprehension of the findings. Therefor, the finding discussed in the text should also be appear in Fig. 4.

*Results (ESL variability)*: The long-range dependence and importance of the 1872 event in sampling ESL data is already demonstrated sufficiently. While the assumption seems obvious to transfer this finding to ESL at the other tide gauges, the authors should comment on this assumption directly for example by determining the Hurst exponent for the Wismar and Warnemünde records as well to prove regional similarities or by referencing other sources discussing the variability.

*Discussion*: Even though EVA not considering historical data in not the focus of the paper, the discussion would benefit from an evaluation of both methods, whenever possible. At least for the Flensburg tide gauge it would be interesting to see, how an EVA of AMAX data, constructed from the hourly record, performs in comparison to the POT-GP method. This would enable more thorough comments on some aspects already mentioned in the paper:

- In the introduction it is mentioned, that the POT-GP method is generally preferred. Can this notion be confirmed for the German Baltic coast, if the length of the record is the same?
- More importantly, the importance of the inclusion of the 1872 in the AMAX record is shown during the analysis to ESL variability at Travemünde. An example of a tide gauge, where systematic records of this event are not available would further highlight this aspect.
- The inclusion of historic data is show to also improve the AMAX-GEV analysis at Travemünde, and improvements to the ESL estimate are found, due to the reduction of uncertainties. Analysis of a second tide gauge would give some indication, whether these improvements are similar at the other tide gauges, whether historic data improves POT-GP and AMAX-GEV analysis in a similar way at each tide gauge, and of the potential improvements, if additional historic data becomes available at the other tide gauges.

*Discussion (1872 event):* The authors briefly discuss the inclusion of the 1872 event and conclude, that this extreme event should be included in the analysis. This conclusion would strongly benefit from further evidence and discussion, for example by showing the estimated return frequency of the event with and without the inclusion of historical data. Further information should be given, which criteria were previously used to call the event an outlier. Are these criteria still valid or does the inclusion of historical data enable a new way of defining an outlier?

Specific comments:

Line 30-32: The mentioned paper of Weiss et al. (2014) uses the concept of homogenous regions for extreme wave height statistics in more offshore areas. Does literature on the feasibility of this method for estimating extreme still water levels exist? I suspect additional challenges posted by the strong influence of coastal bathymetry.

Line 40: Grammatical error and I suggest a more direct phrasing:

> Despite this, several statistical methods exist to …

Line 62: Missing bracket after the reference to Fig. 1.

Line 72: … (hereafter referred as HW200) …

Line 73: … of past observations,  whose accuracy …

Lines 74-75: At least for Mecklenburg-Western Pomerania additional information is available (MLUV, 2012), which mentions the Gumbel distribution to generally result in the best fit. However, the reassessment of the described methodology conducted in 2021 does not yet seem to be published. Nonetheless, I suggest, extending the description of current design practices. Additionally, the

question arises, whether usage of the Gumbel distribution would be a logical addition to the analysis of this paper to enable the most direct comparison to current design practices.

Figure 2: Please provide a point of reference and a definition for the event's magnitude. I presume magnitude refers to the maximum height above MSL.

Figure 5: The description should include the name of the tide gauge to clarify its context.

Line 279: … series of AMAX ESLs at Travemünde …

Line 289-290: A question for clarity: does the historical data in this case only refer to historical data from literature or does it also include "historical" values from the AMAX record outside the 70-year window? As a small bump around 1943 is also visible in the "Sys+Hist" plot as well, I would suspect relatively large influence of the inclusion/exclusion of the 1872 event as well.

Line 387: … Wismar: 63y years …

Line 388: I suggest using the abbreviation HW200 for consistency:

> … estimates for HW200 increase …

References:

MLUV: Ministerium für Landwirtschaft, Umwelt und Verbraucherschutz Mecklenburg-Vorpommern: Regelwerk Küstenschutz Mecklenburg-Vorpommern - Bemessungswasserstand und Referenzhoch-wasserstand, Nr. 2-5, https://www.stalu-mv.de/serviceassistent/download?id=1634742, 2012

---

## Referee Comment (RC3)

The manuscript by MacPherson et al. presents the importance of considering historical information, primarily related to large events, in the extreme sea level estimation. Including such data, instead of treating them as outliers, it can reduce the uncertainties to better plan for coastal adaptation. I find the article to be of very good quality, well written, with a detailed methodology and with a good discussion. However, I do recommend some improvement in the results section.

The findings presented here are not only of great interest to the coastal research community but also to coastal managers. In this sense, my overall impression is positive, and I recommend the publication. I nonetheless do have some minor comments that may help to improve the manuscript by a bit.

**Specific comments:**

Figure 1: Define in Figure 1 where "Kattegat", "Schleswig-Holstein" and "Mecklenburg-Western Pomerania" are located.

Figure 2: Although Figure 2 exemplifies the data series, it does not contain the water level. No scale illustrates the events' magnitude, so the reader cannot visualize the water level. Including the information related to the 1872 event. How large was it? I recommend the authors plot the data series for each site containing the thresholds presented in lines 191 and 192 and the POT and AMAX selected events.

Line 191: The authors justify why they had to select different thresholds for each site but do not give major details. In this sense, explain the criteria for choosing each threshold in more detail.

Line 199: "…AMAX samples are used in lieu of historical measurements for analysis 2,…" What is analysis 2? Also, the authors say "lack of historical information," but the AMAX samples are longer than the historical measurements in Eckernförde and Kiel, for example. Clarify these sentences.

Line 205: If you refer to what can be seen in Figure 2, I cannot see it properly. Figure 2 should be reformulated to better visualization of the events.

Line 228: Why a 70-year moving window?

Line 240: You haven't mentioned HW1000 before. Please include it in the methodology.

Line 245: The changes in the maximum likelihood estimation being negligible for AMAX makes me wonder what the results would be for the other sites that you used POT only. Have you tried to apply AMAX on the other sites with the data you have? Could this be related to the method applied (POT x AMAX)?

Line 251: I believe you are saying 71 years of systematic data, but how many POT events are considered in these 71 years? Is the result difference caused by the amount of sampling or the different techniques? This discussion could be added in the manuscript.

Table 2: Review all numbers referred to in Table 2 in the text. It could be a rounding issue, but some of the numbers presented in the text are slightly different from the ones in the table—for example, the difference shown in line 272 is 47 instead of 48 cm.

Line 273: The authors present differences in HW200 for Wismar and Warnemünde (2 cm and 5 cm, respectively) but then say in line 244 that "the effect of incorporating historical information can only be examined at Travemünde". Where do these results come from?

The authors say the systematic record at Travemünde started in 1949. But there has been a continuous measure of AMAX since 1826. In this case, the systematic data mentioned in Figure 5 are from 1826, right? And what about the syst.+hist one? If historical data are the white circles in Figure 2, in a moving average of 70 years, from 1900 onwards, it would include only one datum, the one from 1872. Is this one event changing that much in the results from using systematic data only when compared to systematic + historical data? Later you explore this better in the discussion, but I believe that Figure 5a needs to be better described in the results section.

**Minor corrections:**

Line 62: Close parenthesis in Figure 1.

Line 71: (MELUND 2022) → (MELUND, 2022)

Lines 71/79: (MLUV 2009) → (MLUV, 2009)

Figure 2 caption: Define AMAX and POT.

Line 80: These data cover only the coastline of Mecklenburg-Western Pomerania except for at Travemünde, which is a special case due to its location at the border of both states. → These data cover the coastline of Mecklenburg-Western Pomerania and also Travemünde, located at the border with Schleswig-Holstein.

Line 85: and → an

Line 171: Define NHN.

Table 2 caption: The first figure… → The first table… Add units.

Figure 3: Define MLE.

Line 257: 54% → 51%

Line 264: Wismar → Warnemünde

Line 279: "…large increases…" Rewrite this sentence since the changes at Travemünde, including historical information, are negligible.

Figure 5: Correct Figure 5 to include all the CI data. Otherwise, the reader cannot see the 2.63m difference mentioned in line 292.

Line 296: Figure 5b.

---

## Author Comment (AC1)

The authors would like to thank the reviewer for their comments. We appreciate the time taken to read through the manuscript and provide specific comments to improve it. We have addressed each comment and listed all changes here. We believe the manuscript has greatly improved thanks to these suggestions.

General comments:

*Data*: The length of most of the hourly records does not seem to match the length of the GESLA3 records. Please provide the data sources used to extend the records.

> Thanks for pointing this out. The analysis was first conducted using only data from GESLA3 but was later changed to include further systematic data provided by Schmidt et al. (2017), which was sourced from the local shipping authorities. We have include this source in Table 1, and updated the statement on data availability:

> > *"Systematic tide-gauge data was sourced from the GESLA-3 database (Haigh et al., 2022) in combination with data from MacPherson et al. (2019), provided by Kelln et al. (2017). The latter is available from local water and shipping authorities along the German Baltic Sea coast upon request."*

*Data preparation and filtering*: A few short comments on the methods of data preparation and their potential influence on the results seem helpful for the reader. Specifically, these questions arise:

While certainly the easiest methods of detrending, extending the linear trend into the first half of the 19[th] century (line 168) seems prone to overestimate sea level rise in extended timespan due to possible acceleration between the 19[th] and 20[th] century. This requires at least a short discussion of the underlying assumptions and their uncertainties and whether other option to fill this gap (e.g. a quadratic trend) were tested.

> This is a good point and we have re-examined our method to determine if there are any large biases due to our choice of extending the MSL trend linearly. At Travemünde and Warnemünde, the largest difference in the MSL signal extended using linear and quadratic fitting is approximately 1 cm. However, at Wismar, a maximum difference of approximately 5 cm is found. While this difference is noteworthy, we do not think it is necessary to perform the analysis again. Indeed, a quadratic trend may still not be the best choice, and examining a range of options would require significant work. However, we have noted these differences in the manuscript and included a short discussion highlighting potential biases caused by this assumption:

> > *"This method may not be the most appropriate due to accelerations in the rate of sea level rise. Although a quadratic trend results in differences of less than 1 cm at Travemünde and Warnemünde, at Wismar a maximum difference of approximately 5 cm is found. The use of a linear trend over a quadratic trend results in an increase to the AMAX samples not covered by the PSMSL data, which in turn leads to a positive bias of the final ESL estimates. However, it is unclear whether the quadratic trend would be better suited to the data, and in combination with the minor differences seen at Travemünde and Warnemünde, a linear trend is considered suitable for our purposes."*

The authors explain sufficiently, why an additional threshold for the selection of the historic events is necessary, but find that no systematic method was able to derive an appropriate threshold for all tide

gauges. They therefore resort to a manual review of data. To understand the process of this selection, a description of the (possibly subjective) criteria for this review would be helpful to the reader.

The selection of the perception threshold is indeed subjective, and is based purely on intuitive decision-making by the authors given the available data. A description of this has been included in the manuscript. We agree that a more thorough description of this process should be included in the paper. We have inserted the following:

*"Given the lack of a clear physical threshold at any of the tested locations (e.g. a sea wall where all exceeding events are recorded), a threshold selection process was conducted based simply on the author's intuitive reasoning. Factors that influenced the selection process include the magnitude and occurrence of ESLs in both the systematic and historical records and the length of the historical record in question. Keeping in mind the assumption that the historical record is exhaustive, and due to the subjective nature of this method, final perception thresholds were set conservatively high at 2.3 m at Flensburg, 2 m at Schleswig, 2.25 m at Eckernförde, 2.25m at Kiel, 2.6 m at Travemünde, 2.25 m at Wismar and 2 m at Warnemünde. Historical ESLs that do not exceed the perception threshold cannot be used in the analysis, and are thus disregarded. These events are highlighted in Figure 2."*

Furthermore, the consequences of this selection remain unclear until much later in the paper, where it is briefly mentioned that only the 1872 event remains at four stations after applying this threshold (line 355-357). This is important information for the reader and should be mentioned alongside the definition of the filter.

It is briefly mentioned in the original manuscript that all historical events that do not exceed the perception threshold are disregarded – "All historical information that lies below the perception threshold was disregarded." (Line 192). To make the effect of this filter clearer to the reader, we have modified Figure 2 to highlight those events which have not been used in the analysis. Further, it appears that only 3 sites were conducted using only the 1872 event as a historical measurement (Eckernförde, Kiel and Warnemünde). We have also corrected this in the manuscript.

[Figure]

**Figure 2.** The extent of data available at all locations. Each circle denotes a sampled ESL with its size proportional to the event's magnitude. All data has been detrended using MSL. Historical events which lie below the perception threshold and are thus disregarded in the final analysis are shown with a black cross.

In this context, it also is unclear which step of data preparation Fig. 2 depicts. Some parts of the data preparation (the extension of the AMAX timeseries) seems to have already been conducted, others (applying the threshold for historical data) not.

This is a good point and we have updated Figure 2 and its caption to inform the reader what data is actually shown (see above).

Results (ESL estimates): The results are presented in an overall clear and easy to understand fashion. Still, an addition to Fig. 4 would highlight the results of the study even more. I suggest to include the AMAX-GEV (Sys.+HI) estimate of Travemünde as well to show that the AMAX estimate benefits from the inclusion as well (even though the interpretation is limited by the availability of only one example). I further suggest adding a figure similar to Fig. 4, but depicting the HW1000 estimates, as supplementary material. These values are already given in Table 2, but I found Fig. 4 to enable a much easier comprehension of the findings. Therefor, the finding discussed in the text should also be appear in Fig. 4.

Thankyou for this comment and we made the suggested changes. Figure 4 now includes the AMAX-GEV (Syst + HI) analysis for Travemünde and we have split the figure into subplots which provide the HW200 and HW1000 estimates:

[Figure]

**Figure 4.** Comparison of **(a)** HW200 and **(b)** HW1000 estimates at all sites. Maximum likelihood estimates are shown as black horizontal lines. 95% credibility intervals are shown as colored bars. Where historical information is included, ESL estimates increase at all sites and credibility intervals are generally reduced. This occurs for both the POT-GP and AMAX-GEV analyses.

Results (ESL variability): The long-range dependence and importance of the 1872 event in sampling ESL data is already demonstrated sufficiently. While the assumption seems obvious to transfer this finding to ESL at the other tide gauges, the authors should comment on this assumption directly for example by determining the Hurst exponent for the Wismar and Warnemünde records as well to prove regional similarities or by referencing other sources discussing the variability.

We agree with this comment and performed the analysis at the two sites mentioned. Our findings have been included in the Manuscript in Section 4.2:

> *"Similarly high Hurst exponents were found at Wismar (0.77) and Warnemünde (0.62). This suggests that there is persistency in the series of ESLs at Travemünde, Wismar and Warnemünde which can be seen as some long-term variability."*

*Discussion*: Even though EVA not considering historical data in not the focus of the paper, the discussion would benefit from an evaluation of both methods, whenever possible. At least for the Flensburg tide gauge it would be interesting to see, how an EVA of AMAX data, constructed from the hourly record, performs in comparison to the POT-GP method. This would enable more thorough comments on some aspects already mentioned in the paper:

In the introduction it is mentioned, that the POT-GP method is generally preferred. Can this notion be confirmed for the German Baltic coast, if the length of the record is the same?

Confirming a better model is difficult as standard techniques such as using Bayesian Information Criteria (BIC) or similar, are not designed to handle two different samples, which is the case when we use two different methods to sample the same dataset. In any case, POT-GP generally performs better at maintaining reasonable uncertainties at the tails, which is supported by Arns et al. (2013) and Wahl et al. (2017). In reality, either choice would provide sufficient results for our study site, but we decided to employ the POT-GP method as we were already forced to use AMAX-GEV for the longer AMAX data set. In this way, we could demonstrate the method for the two most common approaches to EVA. This is an interesting point however, and we have included a comparison of the POT-GP and AMAX-GEV approaches in Appendix A: ESL Sampling:

> *"In this study, we employ two different approaches for sampling ESLs. The first technique, POT is generally preferred in literature for reasons explained in Section 2.2. Wahl et al. (2017) also note that AMAX sampling may result in larger uncertainties at the tails of the distribution when sea level records are short. While we are constrained by the records sourced from MLUV (2012), which are only available as AMAX samples, this is not the case for the high-resolution tide-gauge data. Thus, we decided to employ POT sampling for these records due to their preference in literature, and so that we may demonstrate the use of the Bayesian MCMC EVA method for both POT-GP and AMAX-GEV approaches. To directly compare the two approaches, we also performed an AMAX-GEV analysis using the high-resolution tide-gauge data. Figure A1 shows ESL estimates including 95% credibility intervals estimated at each site using the high-resolution tide-gauge data only. At all sites, the AMAX-GEV method results in larger estimates of ESLs at high return periods. Also of note are the larger uncertainties at the tails of the distribution at all sites except for Warnemünde, which supports the findings of Wahl et al. (2017). In general, the POT-GP appears to produce more reliable results based purely on the reduced uncertainties."*

[Figure]

**Figure A1.** Comparison of POT-GP and AMAX-GEV approaches to the Bayesian MCMC method for estimating ESLs. At each site, estimates of ESLs are based on the high-resolution tide-gauge data, and thus record lengths are the same for both approaches. In general, the AMAX-GEV approach results in higher estimates of ESLs at high return periods and larger uncertainties at the tails of the distributions.

Following a comment from another reviewer, we also considered how this result is affected by the inclusion of historical information:

> *"Next, we considered how these estimates are affected by the addition of historical information. We performed the analysis again, but included historical information with measurement uncertainties given by Jensen et al (2022). Results are shown in Figure A2. As with the first analysis, both the POT and AMAX samples are taken from the high-resolution tide-gauge record. For both POT-GP and AMAX-GEV analysis, the introduction of historical information is beneficial in terms of reduced estimate uncertainties. Interestingly, we find that the AMAX-GEV approach performs better in terms of reduced uncertainties at all sites except Schleswig. Differences in the maximum likelihood estimates between the two analyses are much reduced."*

[Figure]

Figure A2. Comparison of POT-GP and AMAX-GEV approaches to the Bayesian MCMC method for estimating ESLs including historical information. At each site, estimates of ESLs are made using the same high-resolution tide-gauge record in combination with historical information. In contrast to results where historical information is omitted, the AMAX-GEV approach performs somewhat better than the POT-GP approach in terms of reduced uncertainties at the distribution tails. Differences in the maximum likelihood estimates between the two methods are much reduced.

We have also included a paragraph dedicated to this in the Discussion:

*"Large differences exist in the estimates of ESLs made using either the POT-GP or AMAX-GEV approaches. While the incorporation of historical information reduces these differences, it does not provide any insight into which method performs best. Indeed, the POT-GP approach is generally preferred in the literature (Arns et al. 2013, Wahl et al. 2017}, but this does not necessarily apply to the case of the German Baltic Sea coast. We find that when both methods are constrained to the same record length (see Appendix A), the POT-GP method generally performs better with lower uncertainties at the distribution tails. At all sites, the AMAX-GEV provides larger ESL estimates at high return periods. Interestingly, including historical information in the analysis produces a different result, with the AMAX-GEV analysis providing lower uncertainties at high return periods. One possible explanation for this involves the sampling threshold for the POT method. We assume that this threshold is constant for the full duration of historical and systematic observation, following the study by Bulteau et al. (2018), but this may not be the case. Indeed, large differences in results due to the inclusion of historical information suggests this assumption may be false. Thus, an advantage of the AMAX-GEV approach is that no sampling threshold is required. Given a single sea level record with no historical information, we would recommend the POT-GP approach over the AMAX-GEV due to the reduced estimate uncertainties. However, the AMAX-GEV approach may provide more*

*precise results when historical information is available. Where a longer AMAX record is available, such as in this study, the AMAX-GEV approach provides clearly better results due to the increased data."*

More importantly, the importance of the inclusion of the 1872 in the AMAX record is shown during the analysis to ESL variability at Travemünde. An example of a tide gauge, where systematic records of this event are not available would further highlight this aspect.

Thankyou for this comment but we believe that an extra example would not necessarily add much to our findings. A tide-gauge record that does not include 1872 would not allow for such a long period of ESL estimates to be made (as seen in Figure 5(a)). Given the length of the AMAX window used in Figure 5 (a), considering only the latter part of the ESL estimates (1943 onwards) is itself an example of a shorter tide-gauge. Furthermore, direct comparisons between the high-resolution tide-gauge data and AMAX records at Travemünde, Wismar and Warnemünde in Figure 4 show the influence of the 1872 event.

The inclusion of historic data is show to also improve the AMAX-GEV analysis at Travemünde, and improvements to the ESL estimate are found, due to the reduction of uncertainties. Analysis of a second tide gauge would give some indication, whether these improvements are similar at the other tide gauges, whether historic data improves POT-GP and AMAX-GEV analysis in a similar way at each tide gauge, and of the potential improvements, if additional historic data becomes available at the other tide gauges.

Unfortunately, Travemünde is the only site where both a long AMAX record and historical information can be found. At Wismar and Warnemünde, where a long AMAX record is available, no information on historical ESLs exist. Despite this, we have since added a comparison of the AMAX-GEV and POT-GP approaches as an Appendix (Appendix A. see above comment) and find that, in general, there are larger benefits to the AMAX-GEV approach in comparison to the POT-GP approach when historical information is included in EVA.

*Discussion (1872 event):* The authors briefly discuss the inclusion of the 1872 event and conclude, that this extreme event should be included in the analysis. This conclusion would strongly benefit from further evidence and discussion, for example by showing the estimated return frequency of the event with and without the inclusion of historical data. Further information should be given, which criteria were previously used to call the event an outlier. Are these criteria still valid or does the inclusion of historical data enable a new way of defining an outlier?

Thankyou for this point. We have expanded on our description of the 1872 event and included a more thorough discussion on its classification as an outlier. In Section 4.1 we have added a table with its estimated return periods:

*"Including historical information also allows for a more reasonable representation of the 1872 event. Table 3 shows the estimated return period in years of the 1872 event at each site and for each analysis. Given only high-resolution tide-gauge data, return period estimates of 1872 are not realistic, suggesting that the event is an outlier. At travemünde, 1872 is estimated to have a return period of more than 500 billion years. Furthermore, no estimates could be made at Flensburg, Schleswig, Eckernförde or Kiel, as the 1872 value is not defined within the maximum likelihood distributions. At Wismar and Warnemünde, estimated return periods are also high at approximately 4 million*

*years. When historical information is included, return periods of between 700 and 2860 years are assigned to the 1872 event. These estimates are in the same order of magnitude provided by the AMAX-GEV analyses, which include the 1872 event within the systematic data."*

**Table 3.** Return period estimates (years) of the 1872 event at all sites. Estimates that are 'Undefined' represent values which lie outside the range of the distribution. Values that could not be found due to missing data are shown as dashes (-).

| Site | POT-GP (Syst.Only) | POT-GP (Syst. + H.I.) | AMAX-GEV (Syst.Only) | AMAX-GEV (Syst. + H.I.) |
|---|---|---|---|---|
| Flensburg | Undefined | $1.57 \times 10^3$ | - | - |
| Schleswig | Undefined | $2.86 \times 10^3$ | - | - |
| Eckernförde | Undefined | $1.63 \times 10^3$ | - | - |
| Kiel | Undefined | $1.37 \times 10^3$ | - | - |
| Travemünde | $5.59 \times 10^{11}$ | $1.90 \times 10^3$ | $6.80 \times 10^3$ | $5.82 \times 10^3$ |
| Wismar | $3.99 \times 10^6$ | $0.73 \times 10^3$ | $0.90 \times 10^3$ | - |
| Warnemünde | $4.23 \times 10^6$ | $2.29 \times 10^3$ | $6.76 \times 10^3$ | - |

And in the discussion:

*"The largest influence on the results of this study is the event of November 1872, which caused exceptionally high water levels along much of the German Baltic coast. Differences in estimates produced using systematic tide-gauge data and AMAX records are largely due to this event. In fact, 1872 was the only event considered in combination with tide-gauge data at three of the seven sites tested, however changes to the maximum likelihood estimates due to its inclusion are not indifferent to results at other sites where multiple historical events are considered. Due to the exceptional magnitude of the event, it has until recently been treated as an outlier and thus excluded from statistical analyses. Hofstede and Hamann (2022) argue that based on the series of AMAX ESLs at Travemünde, 1872 is indeed an outlier given that it is more than 50% higher than the second largest event. Indeed, Mudersbach and Jensen (2009) assessed the return period of a corrected 1872 event at about 10,000 years. However, they concluded that the event could not be well defined statistically given the limited sample population, and suggest extending the available data using historical information. Jensen et al. (2022) highlight the occurrence of events within historical records of similar magnitudes to 1872. Given these events, Jensen et al. (2022) argue that 1872 should not be considered an outlier and that the systematic records are not sufficiently long to deal with events of these magnitudes.*

*When considering the full historical record at Travemünde in combination with high-resolution tide-gauge data, we assess the return period of the 1872 event to be approximately 1,900 years. Combining the long AMAX record with historical information provides a return period estimate of approximately 5,800 years. Given the length of the historical record at Travemünde (approximately 980 years), and the occurrence of other large ESLs within it (1320, 1625, 1694), we agree with the arguments of Jensen et al. (2022) that the 1872 event should not be considered an outlier, but rather an exceptional realisation of the underlying ESL distribution, and we would recommend for its use in EVA. While sea level records that cover a period that includes 1872 can provide very good ESL estimates using traditional EVA methods, only few sufficiently long records exist along the German Baltic coast (including the AMAX records described herein). Despite*

*this, we show that even short tide-gauge records (approximately 30 years in our case) with one measurement of 1872 can provide similar results. Therefore, reconstructions of past extremes offer valuable information with which to improve EVA."*

*Specific comments:*

*Line 30-32: The mentioned paper of Weiss et al. (2014) uses the concept of homogenous regions for extreme wave height statistics in more offshore areas. Does literature on the feasibility of this method for estimating extreme still water levels exist? I suspect additional challenges posted by the strong influence of coastal bathymetry.*

This method has been applied to estimate extreme still waters with mixed results. I've included two studies where RFA has been applied to still water levels. (Arns et al. 2015, Bardet et al. 2011)

Line 40: Grammatical error and I suggest a more direct phrasing:
Despite this, several statistical methods exist to …
Changed!

Line 62: Missing bracket after the reference to Fig. 1.
Fixed!

Line 72: … (hereafter referred as HW200) …
Fixed!

Line 73: … of past observations, and whose accuracy …
Changed!
Lines 74-75: At least for Mecklenburg-Western Pomerania additional information is available (MLUV, 2012), which mentions the Gumbel distribution to generally result in the best fit. However, the reassessment of the described methodology conducted in 2021 does not yet seem to be published. Nonetheless, I suggest, extending the description of current design practices. Additionally, the question arises, whether usage of the Gumbel distribution would be a logical addition to the analysis of this paper to enable the most direct comparison to current design practices.

I have added this information into Section 2.1. As for the Gumbel distribution, this is a special case of the Generalized Extreme Value distribution when the shape parameter (xi) is equal to 0. As we are using the GEV distribution to model the AMAX data, the Gumbel distribution is included. I have included this information in Section 2.2.:

*"A special case of the GEV distribution occurs when the shape parameter is equal to 0. Here, the distribution becomes a Gumbel distribution, which is mentioned by MLUV (2009) as the best fit for ESLs along the Mecklenburg-West Pomeranian coast."*

Figure 2: Please provide a point of reference and a definition for the event's magnitude. I presume magnitude refers to the maximum height above MSL.

It refers to height above NHN, which I have added into the Figure caption.

Figure 5: The description should include the name of the tide gauge to clarify its context.

Included!

Fixed!

This does indeed include all events above the perception threshold, from both the historical record and the AMAX dataset. The small jump and dive in the Syst. + Hist. line I assume is due to the way systematic and historical information are handled by the Bayesian MCMC method. As the window passes over the 1872 event, it is handled with some level of uncertainty, which was not the case before.

Fixed!

Changed!

References:

MLUV: Ministerium für Landwirtschaft, Umwelt und Verbraucherschutz Mecklenburg-Vorpommern: Regelwerk Küstenschutz Mecklenburg-Vorpommern - Bemessungswasserstand und Referenzhoch-wasserstand, Nr. 2-5, https://www.stalu-mv.de/serviceassistent/download?id=1634742, 2012
Fixed!

---

## Author Comment (AC2)

Thankyou very much to the reviewer for taking the time to read through our manuscript. We have implemented the following changes as per your suggestions!

For Fig. 1, it might be easier to understand if the place names are written on the map.

This change has been made:

[Figure]

For the size of the cycle in Fig. 2, it would be easier to understand by adding the legend of magnitude.

Thankyou for this comment. Please find the updated figure below:

[Figure]

The "exp" in Eqs. (3), (4), and (10) need to be modified to be normal, not italic.

Well spotted! These have been corrected.

Some abbreviations, such as NHN at p. 8 in line 171 and MLE at Fig. 3, are used without explanations. Please check for missing explanations of abbreviations.

Thankyou for this comment. Indeed, some of the explanations have been missed and we have checked the manuscript to correct this.

For Line 192, how much impact does the threshold variation or the number of used historical information have on the extreme value analysis?

Selection of the perception threshold can have a large influence on the final results. If it is set too low, the assumption that the historical records are exhaustive may be false and ESL estimates will be biased downwards. Set too high and some historical events may be disregarded unnecessarily. We performed a sensitivity analysis with a range of perception thresholds and found, that in general, higher perception thresholds result in higher maximum likelihood estimates and vice versa. Although, uncertainties increase as less events are included. Choice of a perception threshold is indeed an important part of this analysis and great care should be taken to set a threshold appropriately high enough to fulfill the exhaustivity assumption, and appropriately low enough to ensure the most amount of data is used. We have expanded the description of how we chose perception thresholds in the manuscript:

*"As mentioned in Section 3.1, it is a necessary condition that the available historical information is exhaustive above a perception threshold. That is, the only events which have exceeded the perception threshold for the duration of historical observation exist within the historical record. Therefore, the perception threshold should be set high enough to ensure this assumption is true. At first, a systematic approach to setting a perception threshold was attempted based on the systematic data and the period of historical information. Here, ESLs were estimated using systematic data only for return periods dependent on the number of historical events available and the length of the historical record. For example, the perception threshold might be set to a height equivalent to a 1-in-100 year event, where a 200 year long historical record is available which contains 2 events. However, due to large differences in the magnitude of systematic and historical observations, relying on the systematic data alone was not sufficient, and no one method could be applied at all sites. Instead, perception thresholds were chosen on a site-by-site basis, using all available data for each case.*

*Given the lack of a clear physical threshold at any of the tested locations (e.g. a sea wall where all exceeding events are recorded), a threshold selection process was conducted based simply on the author's intuitive reasoning. Factors that influenced the selection process include the magnitude and occurrence of ESLs in both the systematic and historical records and the length of the historical record in question. Keeping in mind the assumption that the historical record is exhaustive, and due to the subjective nature of this method, final perception thresholds were set*

*conservatively high at 2.3 m at Flensburg, 2 m at Schleswig, 2.25 m at Eckernförde, 2.25m at Kiel, 2.6 m at Travemünde, 2.25 m at Wismar and 2 m at Warnemünde. Historical ESLs that do not exceed the perception threshold cannot be used in the analysis, and are thus disregarded. These events are highlighted in Figure 2."*

---

## Author Comment (AC3)

The authors would like to thank the reviewer for their time in reading through our manuscript. We are happy to hear that the study was well received and have made some changes as per your suggestions. Here we list those changes.

[Specific comments]

Figure 1: Define in Figure 1 where "Kattegat", "Schleswig-Holstein" and "Mecklenburg-Western-Pomerania" are located.

> We have updated the figure to include the names of the states, and included a description of their border in the figure caption:

[Figure]

Figure 2: Although Figure 2 exemplifies the data series, it does not contain the water level. No scale illustrates the events' magnitude, so the reader cannot visualize the water level. Including the information related to the 1872 event. How large was it? I recommend the authors plot the data series for each site containing the thresholds presented in lines 191 and 192 and the POT and AMAX selected events.

> Thankyou for this comment. We decided to keep the current figure as it is used more to depict the extent of data rather than show precisely the water levels. We have added a scale to the figure so that some idea regarding the magnitude of the events can be interpreted. When we plot the data at all seven sites, it is difficult to discern the differences in record lengths, which is the intent of this figure. We have also mentioned the height of the 1872 ESL event in Section 2.1:
>
> > "The resulting ESLs along the German coast remain the highest on record, registering approximately 3.4 m above mean sea level (MSL) at Travemünde."

[Figure]

In addition to these changes, we have included an extra figure which shows the AMAX record at Travemünde, highlighting the exceptional nature of the 1872 event:

[Figure]

Line 191: The authors justify why they had to select different thresholds for each site but do not give major details. In this sense, explain the criteria for choosing each threshold in more detail.

We have included a more thorough description of the threshold selection process:

> "As mentioned in Section 3.1, it is a necessary condition that the available historical information is exhaustive above a perception threshold. That is, the only events which have exceeded the perception threshold for the duration of historical observation exist within the historical record. Therefore, the perception threshold should be set high enough to ensure this assumption is true. At first, a systematic approach to setting a perception threshold was attempted based on the systematic data and the period of historical information. Here, ESLs were estimated using systematic data only for return periods dependent on the number of historical events available and the length of the historical record. For example, the perception threshold might be set to a height equivalent to a 1-in-100 year event, where a 200 year long historical record is available

*which contains 2 events. However, due to large differences in the magnitude of systematic and historical observations, relying on the systematic data alone was not sufficient, and no one method could be applied at all sites. Instead, perception thresholds were chosen on a site-by-site basis, using all available data for each case.*

*Given the lack of a clear physical threshold at any of the tested locations (e.g. a sea wall where all exceeding events are recorded), a threshold selection process was conducted based simply on the author's intuitive reasoning. Factors that influenced the selection process include the magnitude and occurrence of ESLs in both the systematic and historical records and the length of the historical record in question. Keeping in mind the assumption that the historical record is exhaustive, and due to the subjective nature of this method, final perception thresholds were set conservatively high at 2.3 m at Flensburg, 2 m at Schleswig, 2.25 m at Eckernförde, 2.25m at Kiel, 2.6 m at Travemünde, 2.25 m at Wismar and 2 m at Warnemünde. Historical ESLs that do not exceed the perception threshold cannot be used in the analysis, and are thus disregarded. These events are highlighted in Figure 2."*

Line 199: "…AMAX samples are used in lieu of historical measurements for analysis 2,…" What is analysis 2? Also, the authors say "lack of historical information," but the AMAX samples are longer than the historical measurements in Eckernförde and Kiel, for example. Clarify these sentences.

We describe four analyses in the previous sentence: "Depending on data availability at each site, we perform four separate analyses using: 1) POT samples only, 2) POT samples with historical measurements, 3) AMAX samples only, and 4) AMAX samples with historical measurements." Analysis 2 refers to the 'POT samples with historical measurements' analysis. The statement "lack of historical information" applies only to the sites at Wismar and Warnemünde, not to Eckernförde or Kiel. Because there are no historical records of ESL at Wismar and Warnemünde, and given we are only considering POT samples from the high-resolution tide-gauge data, we use events from the AMAX record as historical events. We have rephrased the sentence to make this more clear.

> *"At Wismar and Warnemünde, as no historical records are available, ESLs from the AMAX records are used in lieu of historical measurements for the second analysis."*

Line 205: If you refer to what can be seen in Figure 2, I cannot see it properly. Figure 2 should be reformulated to better visualization of the events.

Thankyou for this comment. Indeed, it is difficult to see any reduction in ESL values in the AMAX record from Figure 2. We have included a new figure (Figure 3: see above) which provides the AMAX values at Travemünde.

Line 228: Why a 70-year moving window?

A 70 year moving window was chosen based on the length of the high-resolution tide-gauge record, which is 71 years in length. This should be stated in the manuscript, thanks for highlighting this omission. We have added the following into Section 3.4:

> *"A window size of 70 years was chosen to match the length of the high-resolution tide-gauge record at Travemünde (71 years in length)."*

Line 240: You haven't mentioned HW1000 before. Please include it in the methodology.

Like HW200, this refers to an extreme sea level with a return period of 1000 years. We have added in brackets the meaning of this term to clarify this point.

*"HW1000 (1-in-1000 year ESL event)"*

Line 245: The changes in the maximum likelihood estimation being negligible for AMAX makes me wonder what the results would be for the other sites that you used POT only. Have you tried to apply AMAX on the other sites with the data you have? Could this be related to the method applied (POT x AMAX)?

Thank you for this comment, this is an interesting point which we did not consider. Addressing the comments of another reviewer we performed the Bayesian MCMC analysis at each site using only the high-resolution tide-gauge records. Therefore, we could compare the performance of the POT and AMAX sampling using the same length of data. This was added to the manuscript as an Appendix:

*"Appendix A: ESL sampling*

*In this study, we employ two different approaches for sampling ESLs. The first technique, POT is generally preferred in literature for reasons explained in Section 2.2. Wahl et al. (2017) also note that AMAX sampling may result in larger uncertainties at the tails of the distribution when sea level records are short. While we are constrained by the records sourced from MLUV (2012), which are only available as AMAX samples, this is not the case for the high-resolution tide-gauge data. Thus, we decided to employ POT sampling for these records due to their preference in literature, and so that we may demonstrate the use of the Bayesian MCMC EVA method for both POT-GP and AMAX-GEV approaches. To directly compare the two approaches, we also performed an AMAX-GEV analysis using the high-resolution tide-gauge data. Figure A1 shows ESL estimates including 95% credibility intervals estimated at each site using the high-resolution tide-gauge data only. At all sites, the AMAX-GEV method results in larger estimates of ESLs at high return periods. Also of note are the larger uncertainties at the tails of the distribution at all sites except for Warnemünde, which supports the findings of Wahl et al. (2017). In general, the POT-GP appears to produce more reliable results given the same record duration based purely on the reduced uncertainties.*

[Figure]

Figure A1. Comparison of POT-GP and AMAX-GEV approaches to the Bayesian MCMC method for estimating ESLs. At each site, estimates of ESLs are made using high-resolution tide-gauge data only. In general, the AMAX-GEV approach results in higher estimates of ESLs at high return periods and larger uncertainties at the tails of the distribution.

As per your suggestion, we also repeated this analysis including historical information and found that in general, the AMAX-GEV analysis outperforms the POT-GP analysis:

*"Next, we considered how these estimates are affected by the addition of historical information. We performed the analysis again, but included historical information with measurement uncertainties given by Jensen et al (2022). Results are shown in Figure A2. As with the first analysis, both the POT and AMAX samples are taken from the high-resolution tide-gauge record. For both POT-GP and AMAX-GEV analysis, the introduction of historical information is beneficial in terms of reduced estimate uncertainties. Interestingly, we find that the AMAX-GEV approach performs better in terms of reduced uncertainties at all sites except Schleswig. Differences in the maximum likelihood estimates between the two analyses are much reduced."*

[Figure]

Figure A2. Comparison of POT-GP and AMAX-GEV approaches to the Bayesian MCMC method for estimating ESLs including historical information. At each site, estimates of ESLs are made using the same high-resolution tide-gauge record in combination with historical information. In contrast to results where historical information is omitted, the AMAX-GEV approach performs somewhat better than the POT-GP approach in terms of reduced uncertainties at the distribution tails. Differences in the maximum likelihood estimates between the two methods are much reduced.

We have also included a paragraph dedicated to this in the Discussion:

*"Large differences exist in the estimates of ESLs made using either the POT-GP or AMAX-GEV approaches. While the incorporation of historical information reduces these differences, it does not provide any insight into which method performs best. Indeed, the POT-GP approach is generally preferred in the literature (Arns et al. 2013, Wahl et al. 2017}, but this does not necessarily apply to the case of the German Baltic Sea coast. We find that when both methods are constrained to the same record length (see Appendix A), the POT-GP method generally performs better with lower uncertainties at the distribution tails. At all sites, the AMAX-GEV provides larger ESL estimates at high return periods. Interestingly, including historical information in the analysis produces a different result, with the AMAX-GEV analysis providing lower uncertainties at high return periods. One possible explanation for this involves the sampling threshold for the POT method. We assume that this threshold is constant for the full duration of historical and systematic observation, following the study by Bulteau et al. (2018), but this may not be the case. Indeed, large differences in results due to the inclusion of historical information suggests this assumption may be false. Thus, an advantage of the AMAX-GEV approach is that no sampling threshold is required. Given a single sea level record with no historical information, we would recommend the POT-GP approach over the AMAX-GEV due to the reduced estimate uncertainties. However, the AMAX-GEV approach may provide more*

*precise results when historical information is available. Where a longer AMAX record is available, such as in this study, the AMAX-GEV approach provides clearly better results due to the increased data."*

Line 251: I believe you are saying 71 years of systematic data, but how many POT events are considered in these 71 years? Is the result difference caused by the amount of sampling or the different techniques? This discussion could be added in the manuscript.

The number of POT samples depends on the selected threshold. This is a very important part of the POT approach and care must be taken to select an appropriate threshold. This is described in Section 3.3. If a threshold is set too low, more samples will be allowed but this may bias the analysis by introducing non-extreme events. In our case, there are 119 ESL events within the 71 years of data at Travemünde. The difference between the results of the AMAX and POT analyses when considering systematic data only is indeed due to the large difference in record lengths. There is some differences due to the method employed, but this is a minor consideration when the records are substantially different. A comparison of the two methods using the same observational record has been conducted and included in the manuscript as Appendix A (see above comment).

Table 2: Review all numbers referred to in Table 2 in the text. It could be a rounding issue, but some of the numbers presented in the text are slightly different from the ones in the table—for example, the difference shown in line 272 is 47 instead of 48 cm.

This is indeed due to rounding. All figures have been rounded to the nearest cm in both the table and in the text. When we calculated differences between the figures, we only performed the rounding after the operation was completed to avoid double rounding. This does result in some discrepancies between the table and the text, but is also the most correct method of presenting the results. We have added a note to the caption of Table 2 that all figures have been rounded to the nearest cm.

Line 273: The authors present differences in HW200 for Wismar and Warnemünde (2 cm and 5 cm, respectively) but then say in line 244 that "the effect of incorporating historical information can only be examined at Travemünde". Where do these results come from?

The earlier statement is regarding the AMAX-GEV analysis only. As no historical records are available for Wismar and Warnemünde, it is not possible to conduct the AMAX-GEV analysis with historical information at these sites. The statement on line 273 refers to the POT-GP analysis. As mentioned earlier, when conducting the POT-GP analysis at Wismar and Warnemünde, we use AMAX samples as historical events due to the lack of true historical records. Confusion is due to the poor wording on line 269: "Looking only at the sites where long AMAX records are available (Travemünde, Wismar and Warnemünde), we see much better agreement between POT-GP and AMAX-GEV analyses when historical information is considered" We have updated this sentence to provide clarification:

*"Looking only at the sites where long AMAX records are available (Travemünde, Wismar and Warnemünde), we see much better agreement between the POT-GP analysis*

*including historical information and the AMAX-GEV analysis of systematic data only,*
*despite the significantly shorter tide-gauge records."*

The authors say the systematic record at Travemünde started in 1949. But there has been a continuous measure of AMAX since 1826. In this case, the systematic data mentioned in Figure 5 are from 1826, right? And what about the syst.+hist one? If historical data are the white circles in Figure 2, in a moving average of 70 years, from 1900 onwards, it would include only one datum, the one from 1872. Is this one event changing that much in the results from using systematic data only when compared to systematic + historical data? Later you explore this better in the discussion, but I believe that Figure 5a needs to be better described in the results section.

Thanks for the comment. Firstly, we realise that our naming of the high-resolution tide-gauge record and the AMAX record has introduced some confusion. Although the AMAX record consists of both systematically measured ESLs and historical information, their compilation provides a systematic record of annual maxima sea levels which we can use as systematic data. From line 78, we state that "the longest systematic record was installed at Travemünde at the end of 1949". This refers to the high-resolution tide-gauge record and we have updated the manuscript to clarify this point:

*"The longest high-resolution (hourly sampling) systematic record was installed at*
*Travemünde at the end of 1949."*

Yes, Figure 5 shows an analysis of the AMAX record which begins in 1826 and is considered to be a systematic record given its compilation. The Syst + H.I. plot from Figure 5a shows ESL estimates from a 70 year window of AMAX values including all available historical information (not just those values within the 70 year window). This is described in Section 3.4 (lines 227-230) but we have added the following sentence to clarify our methodology:

*"For the second case, all available historical information is used, even those events which*
*occur outside of the 70-year window."*

The effect of 1872 on ESL estimates is shown in Figure 5. When considering systematic data only (70 years of AMAX ESLs), we can see a large decrease in ESL estimates at 1943 where 1872 is no longer considered (Figure 5a blue). By including historical information, 1872 and other large historical events continue to influence our estimates which remain relatively steady over the period considered (Figure 5a).